# A novel monomeric amyloid β-activated signaling pathway regulates brain development via inhibition of microglia

Hyo Jun Kwon, Devi Santhosh, Zhen Huang*

Departments of Neurology and Neuroscience, University of Wisconsin-Madison, Madison, United States

## eLife Assessment

The study describes a link between beta-amyloid monomers, regulation of microglial activity and assembly of neocortex during development. It brings **valuable** findings that have theoretical and practical implications in the field of neuronal migration, neuronal ectopia and type II lissencephaly. Unfortunately, the evidence is **incomplete** and the manuscript would benefit from additional experiments to clarify the relationship between Ric8a and APP and bolster the findings.

*For correspondence:
zhuang3@wisc.edu

Competing interest: The authors declare that no competing interests exist.

**Abstract** Amyloid β (Aβ) forms aggregates in the Alzheimer's disease brain and is well known for its pathological roles. Recent studies show that it also regulates neuronal physiology in the healthy brain. Whether Aβ also regulates glial physiology in the normal brain, however, has remained unclear. In this article, we describe the discovery of a novel signaling pathway activated by the monomeric form of Aβ in vitro that plays essential roles in the regulation of microglial activity and the assembly of neocortex during mouse development in vivo. We find that activation of this pathway depends on the function of amyloid precursor and the heterotrimeric G protein regulator Ric8a in microglia and inhibits microglial immune activation at transcriptional and post-transcriptional levels. Genetic disruption of this pathway during neocortical development results in microglial dysregulation and excessive matrix proteinase activation, leading to basement membrane degradation, neuronal ectopia, and laminar disruption. These results uncover a previously unknown function of Aβ as a negative regulator of brain microglia and substantially elucidate the underlying molecular mechanisms. Considering the prominence of Aβ and neuroinflammation in the pathology of Alzheimer's disease, they also highlight a potentially overlooked role of Aβ monomer depletion in the development of the disease.

## Introduction

Aβ, a core component of amyloid plaques in the Alzheimer's disease brain, is well known to form oligomers under disease conditions. Studies have shown that the oligomers formed by Aβ are highly toxic, with wide-ranging effects including inhibition of neurotransmitter release, depletion of synaptic vesicle pools, disruption of postsynaptic organization and function, and impairment of multiple forms of synaptic plasticity (*Gulisano et al., 2018*; *He et al., 2019*; *Kim et al., 2013*; *Laurén et al., 2009*; *Lazarevic et al., 2017*; *Parodi et al., 2010*; *Puzzo et al., 2008*; *Shankar et al., 2008*; *Walsh et al., 2002*; *Yang et al., 2015*; *Zott et al., 2019*). These effects likely significantly underpin the pathogenic role of Aβ in Alzheimer's disease and contribute to neuron loss and cognitive decline in patients. Besides its pathological roles, recent studies show that Aβ is also produced in the healthy brain by neurons in a neural activity-dependent manner and regulates the normal physiology of neurons

(*Cirrito et al., 2005*; *Fogel et al., 2014*; *Galanis et al., 2021*; *Garcia-Osta and Alberini, 2009*; *Gulisano et al., 2018*; *Gulisano et al., 2019*; *Morley et al., 2010*; *Palmeri et al., 2017*; *Puzzo et al., 2008*; *Zhou et al., 2022*). For example, consistent with studies showing that Aβ monomers and low-molecular-weight oligomers positively regulate synaptic function and plasticity, administration of these molecules in vivo has been found to improve learning and memory in animals (*Fogel et al., 2014*; *Garcia-Osta and Alberini, 2009*; *Gulisano et al., 2018*; *Gulisano et al., 2019*; *Morley et al., 2010*; *Palmeri et al., 2017*; *Puzzo et al., 2008*). Furthermore, recent studies have shown that Aβ monomers directly promote synapse formation and function and homeostatic plasticity, processes crucial to normal cognitive function (*Galanis et al., 2021*; *Kamenetz et al., 2003*; *Zhou et al., 2022*). Together, these findings have provided crucial insights into the physiological roles that Aβ plays in regulating normal neuronal function in the brain. However, it remains unclear if Aβ also regulates the physiology of glia, nonneuronal cells that also play important roles in normal brain function.

Microglia and astrocytes, two of the major glial cell types in the brain, are known to play critical roles in the normal development, function, and plasticity of the brain circuitry (*Barres, 2008*; *Schafer and Stevens, 2015*). They coordinately regulate, among others, the spatiotemporally specific expression of immune cytokines in the brain that regulate numerous processes of brain circuit development, function, and plasticity (*Zipp et al., 2023*). For example, in the thalamus, a key relay station in the visual pathway, populations of astrocytes have been found to activate the expression of interleukin-33 in a neural activity-dependent manner, induce activity-dependent elimination of supernumerary synapses, and promote the maturation of the visual circuitry in early postnatal life (*Vainchtein et al., 2018*). In the adult hippocampus, in contrast, astrocytes have been found to activate the expression of interleukin-33 under neuronal activity blockade and induce homeostatic synaptic plasticity that maintains circuit activity balance (*Wang et al., 2021*). In the striatum and the neocortex, not only have astrocytes but also have microglia been observed to activate the expression of TNFα (Tumor Necrosis Factor α) upon changes in neural circuit activity and induce homeostatic synaptic plasticity that dampens circuit perturbation (*Heir et al., 2024*; *Lewitus et al., 2016*; *Stellwagen and Malenka, 2006*). In the clinic, the induction of microglial release of cytokines such as TNFα also underpins the application of repetitive transcranial magnetic stimulation, a noninvasive brain stimulation technique frequently used to induce cortical plasticity and treat pharmaco-resistant depression (*Eichler et al., 2023*). In neurodegenerative diseases such as Alzheimer's disease, glial activation, and brain cytokine elevation are also key pathologic factors in disease development (*Colonna and Butovsky, 2017*; *Patani et al., 2023*). Furthermore, elevated TNFα expression by microglia also underlies interneuron deficits and autism-like phenotype linked to maternal immune activation (*Yu et al., 2022*). Thus, the precise regulation of glial cytokine expression in the brain plays a key role in the normal development and function of the brain and its dysregulation is linked to common neurodevelopmental and neurodegenerative diseases. However, how glial cytokine expression is mechanistically regulated by cell–cell communication in the brain have remained largely unknown.

In this article, we report the discovery of a novel microglial signaling pathway activated in vitro by Aβ, the neuron-produced peptide at the center of Alzheimer's disease, that plays a crucial role in precisely regulating the levels of microglial cytokine expression and activity and ensuring the proper assembly of neuronal laminae during cerebral cortex development. We first came across evidence for this pathway in our study of the function of *Ric8a*. *Ric8a* encodes a guanine nucleotide exchange factor (GEF) and molecular chaperone for several classes of heterotrimeric G proteins, which become severely destabilized upon *Ric8a* loss of function (*Gabay et al., 2011*; *Papasergi-Scott et al., 2018*; *Tall et al., 2003*). We found that deletion of *Ric8a* during cortical development resulted in cortical basement membrane degradation, neuronal ectopia, and laminar disruption. However, unlike in classic models of cobblestone lissencephaly, these phenotypes resulted not from *Ric8a* deficiency in brain neural cell types, but from deficiency in microglia. Ric8a-regulated Gα proteins are known to bind to the cytoplasmic domain of the amyloid precursor protein (APP) and mediate key branches of APP signaling in several cell types (*Fogel et al., 2014*; *Milosch et al., 2014*; *Nishimoto et al., 1993*; *Ramaker et al., 2013*). The *Ric8a* cortical phenotypes also resemble those in triple or double mutants of APP family and pathway genes (*Guénette et al., 2006*; *Herms et al., 2004*), suggesting functional interactions. Indeed, we found that *App* deficiency in brain microglia also underpins ectopia formation in *App* family gene mutants. Furthermore, we found that APP and Ric8a form a pathway in microglia

that is specifically activated by the monomeric form of Aβ and that this pathway normally inhibits the transcriptional and post-transcriptional expression of immune cytokines by microglia.

## Results

### Cortical ectopia in *Ric8a:Emx1-Cre* mutants results from non-neural deficiency

To study of the function of *Ric8a*, a GEF as well as molecular chaperone for Gα proteins (*Gabay et al., 2011*; *Papasergi-Scott et al., 2018*; *Tall et al., 2003*), in neocortical development, we deleted a conditional *Ric8a* allele (*Ma et al., 2012*; *Ma et al., 2017*) using *Emx1-Cre*, a *Cre* line designed to target dorsal forebrain neural progenitors in mice (*Gorski et al., 2002*). We found it result in ectopia formation exclusively in the lateral cortex of the perinatal mutant brain (*Figure 1a–d*). Birth-dating showed that the ectopia consisted of both early- and late-born neurons (*Figure 1—figure supplement 1*). Consistent with this observation, neurons in the ectopia also stained positive for both Ctip2 and Cux1, genes specific to lower- and upper-layer neurons, respectively. Interestingly, in cortical areas without ectopia, radial migration of early- and late-born neurons appeared largely normal as shown by birth-dating as well as Cux1 and Ctip2 staining (*Figure 1—figure supplement 2*). This suggests that cell-autonomous defects in neurons are unlikely the cause of the ectopia. At E16.5, clear breaches in the pial basement membrane of the developing cortex were already apparent (*Figure 1—figure supplement 3*). However, unlike classic models of cobblestone lissencephaly, where radial glial fibers typically retract, radial glial fibers in *ric8a* mutants instead extended beyond the breaches. This argues against radial glial cell adhesion defects since they would be predicted to retract. Furthermore, in areas without ectopia, we also observed normal localization of Cajal–Retzius cells, expression of Reelin, and splitting of the preplate, arguing against primary defects in Cajal–Retzius cells. In cobblestone lissencephaly, studies show that ectopia result from primary defects in radial glial maintenance of the pial basement membrane (*Beggs et al., 2003*; *Graus-Porta et al., 2001*; *Moore et al., 2002*; *Satz et al., 2010*). In *Ric8a* mutants, we observed large numbers of basement membrane breaches at E14.5, almost all associated with ectopia (*Figure 1—figure supplement 4*). In contrast, at E13.5, although we also observed significant numbers of breaches, none was associated with ectopia. This indicates that basement membrane breaches similarly precede ectopia in *Ric8a* mutants. However, at E12.5, despite a complete lack of basement membrane breaches, we observed increased numbers of laminin-positive debris across the lateral cortex, both beneath basement membrane segments with intact laminin staining and beneath segments with disrupted laminin staining, the latter presumably sites of future breach (*Figure 1—figure supplement 5*). As a major basement membrane component, the increased amounts of laminin debris suggest increased degradative activity within the developing cortex. Thus, these results indicate that excessive basement membrane degradation, but not defective maintenance, is likely a primary cause of cortical ectopia in *Ric8a* mutants.

To determine the cell type(s) genetically responsible for cortical basement membrane degradation and ectopia in *Ric8a* mutants, we employed a panel of *Cre* lines (*Figure 1e–h'*). To target Cajal–Retzius cells, we employed *Wnt3a-Cre* (*Yoshida et al., 2006*) but found *ric8a* deletion using *Wnt3a-Cre* did not result in ectopia. To target postmitotic excitatory and inhibitory neurons, we employed *Nex-Cre* (*Goebbels et al., 2006*) and *Dlx5/6-Cre* (*Stenman et al., 2003*), respectively, but similarly found neither result in ectopia. These results point to *Ric8a* requirement in cell types other than post-mitotic neurons. To test the involvement of neural progenitors, we employed *Nestin-Cre* (*Graus-Porta et al., 2001*). Previous studies show that deletion of *β1* integrin (*Itgb1*) and related genes by *Emx1-Cre* and *Nestin-Cre* results in similar ectopia phenotypes (*Belvindrah et al., 2006*; *Graus-Porta et al., 2001*; *Huang et al., 2006*; *Niewmierzycka et al., 2005*). To our surprise, deletion of *Ric8a* by *Nestin-Cre* did not result in ectopia (*Ma et al., 2017*; *Figure 1g, g'*). Since *Nestin-Cre*-mediated deletion in neural progenitors is inherited by post-mitotic neurons and astrocytes, this indicates that the combined deletion of *Ric8a* from all these cell types does not lead to ectopia. The onset of *Nestin-Cre* expression is, however, developmentally slightly later than that of *Emx1-Cre* (*Gorski et al., 2002*). To assess the potential contribution of this temporal difference, we employed *Foxg1-Cre*, a *Cre* line expressed in forebrain neural progenitors starting from E10.5 (*Hébert and McConnell, 2000*). We found that *Ric8a* deletion using *Foxg1-Cre* still failed to produce ectopia (*Figure 1h, h'*). Thus, these results strongly

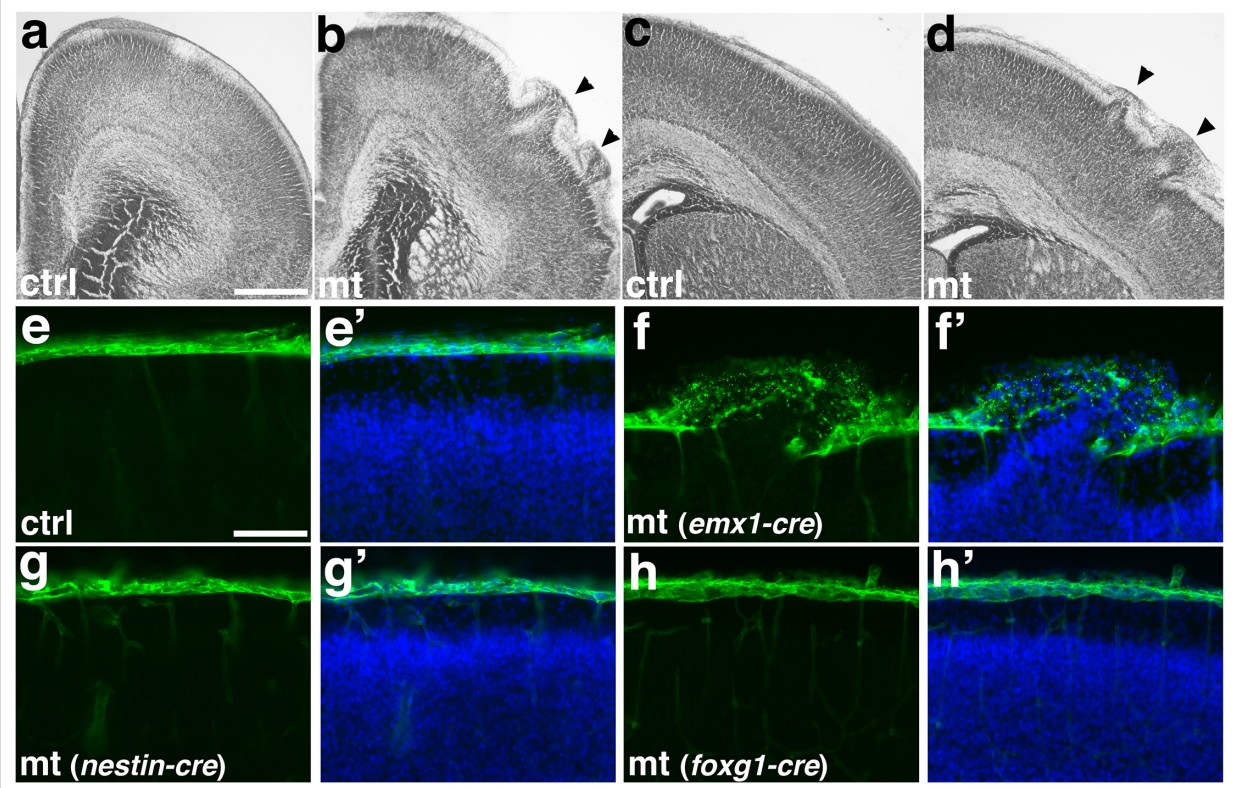

**Figure 1.** Deletion of *Ric8a* using *Emx1-Cre* results in cortical ectopia due to non-neural deficits. (**a–d**) Nissl staining of control (ctrl, **a, c**) and mutant (mt, **b, d**) anterior motor (**a, b**) and posterior somatosensory (**c, d**) cortex at P0. (**e, e'**) Laminin (LN, in green) and nuclear (4′,6-diamidino-2-phenylindole-DAPI, in blue) staining of control cortices at P0. A continuous basement membrane is observed at the pia, beneath which cells are well organized in the cortical wall. (**f, f'**) Staining of *Ric8a:Emx1-Cre* mutant cortices at P0. Basement membrane breach and neuronal ectopia are observed following *Ric8a* deletion by *Emx1-Cre*, a *Cre* line expressed in cortical radial glial progenitors beginning at E10.5. (**g, g'**) Staining of *Ric8a:Nestin-Cre* mutant cortices at P0. No obvious basement membrane breach or neuronal ectopia is observed following *Ric8a* deletion by *Nestin-Cre*, a *Cre* line expressed in cortical progenitors beginning around E12.5. (**h, h'**) Staining of *Ric8a:Foxg1-Cre* mutant cortices at P0. No obvious basement membrane breach or neuronal ectopia is observed following *Ric8a* deletion by *Foxg1-Cre*, a *Cre* line expressed in forebrain neural progenitors from E9.0. Scale bars, 640 μm for (**a, b**), 400 μm for (**c, d**), and 100 μm for (**e–h'**).

The online version of this article includes the following figure supplement(s) for figure 1:

**Figure supplement 1.** Birth-dating of early- and late-born neurons in *Ric8a:Emx1-Cre* mutant cortices.

**Figure supplement 2.** Lamina-specific neuronal markers are normal outside ectopia in *Ric8a:Emx1-Cre* mutant cortices.

**Figure supplement 3.** Neuronal ectopia in *Ric8a:Emx1-Cre* mutants result from pial basement membrane breach during embryogenesis.

**Figure supplement 4.** Basement membrane breaches precede neuronal ectopia in *Ric8a:Emx1-Cre* mutant cortices.

**Figure supplement 5.** Signs of basement membrane degradation before breach formation at E12.5.

**Figure supplement 6.** Cortical radial glial identity and proliferation are unaffected in *Ric8a:Emx1-Cre* mutants.

**Figure supplement 7.** Wnt pathway activity is normal in *Ric8a:Emx1-Cre* mutant cortices.

argue against the interpretation that *Ric8a* deficiency in neural cell lineages is responsible for basement membrane degradation and ectopia in *Ric8a* mutants.

During embryogenesis, the neural tube undergoes epithelial–mesenchymal transition giving rise to neural crest cells (*Leathers and Rogers, 2022*). This process involves region-specific basement membrane breakdown that resembles the *Ric8a* mutant phenotype. To determine if ectopic epithelial–mesenchymal transition plays a role, we examined potential changes in neuro-epithelial cell fates in the mutant cortex. We found that cortical neural progenitors expressed Pax6, Nestin, and Vimentin normally (*Figure 1—figure supplement 6*). Cell proliferation in the ventricular zone was also normal. Furthermore, although *Ric8a* regulates asymmetric cell division in invertebrates (*Afshar et al., 2004*; *Couwenbergs et al., 2004*; *David et al., 2005*; *Hampoelz et al., 2005*; *Wang et al., 2005*), we

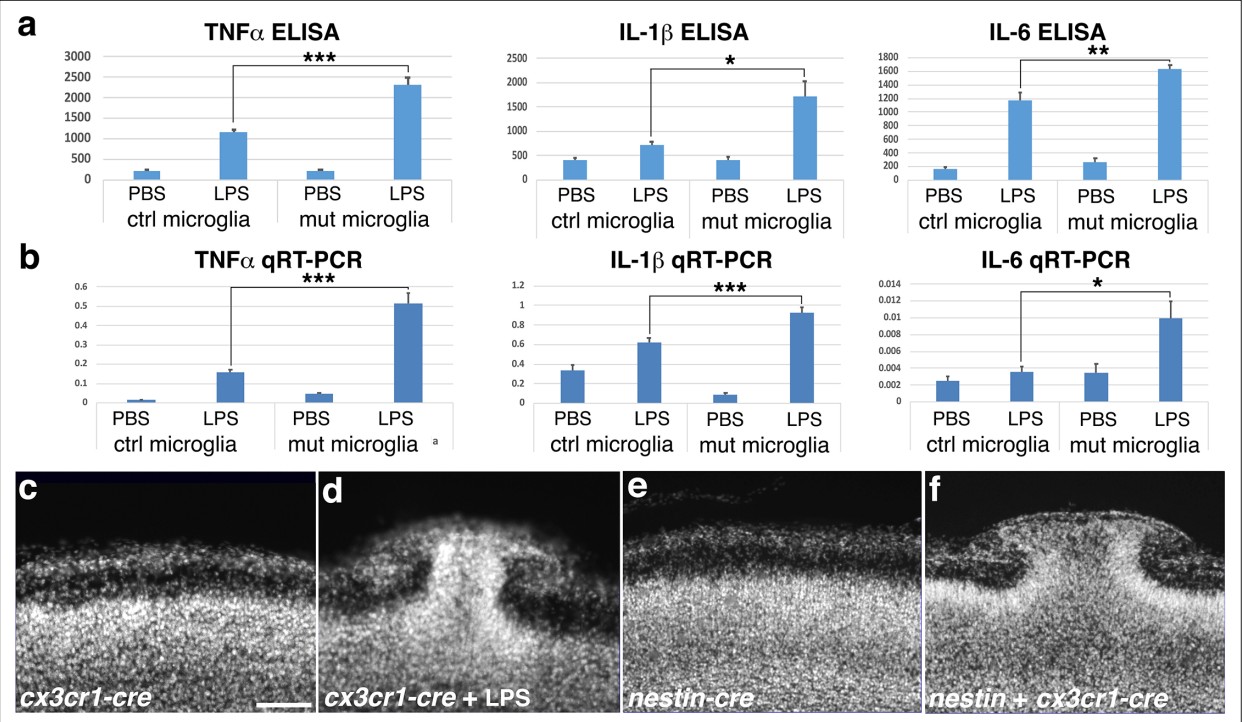

**Figure 2.** *Ric8a* deficiency in microglia is responsible for cortical ectopia. (**a**) TNFα, IL-1β, and IL-6 secretion (pg/ml) in control and *Ric8a:Cx3cr1-Cre* mutant microglia following lipopolysaccharide (LPS) stimulation. *p < 0.05; **p < 0.01; ***p < 0.001; n = 6–8 each group. (**b**) TNFα, IL-1β, and IL-6 mRNA expression in control and *Ric8a:Cx3cr1-Cre* mutant microglia following LPS stimulation. *p < 0.05; ***p < 0.001; n = 5–6 each group. Nuclear (DAPI, in gray) staining of *Ric8a:Cx3cr1-Cre* mutant cortices at P0 in the absence (**c**) or presence (**d**) of LPS treatment during embryogenesis. Nuclear (DAPI, in gray) staining of *Ric8a:Nestin-Cre* single *cre* (**e**) and *Ric8a:Nestin-Cr+Cx3cr1-Cre* double *Cre* (**f**) mutant cortices at P0. Scale bar in (**c**), 100 μm for (**c–f**).

The online version of this article includes the following source data and figure supplement(s) for figure 2:

**Source data 1.** Excel files for control and Ric8a mutant microglia ELISA and qRT-PCR analysis.

**Figure supplement 1.** *Emx1-Cre* is active in microglia.

**Figure supplement 2.** Gαi protein is severely depleted from *Ric8a:Emx1-Cre* mutant cortices.

**Figure supplement 2—source data 1.** Western blot analysis of Gαi levels in E13.5 and P0 brains.

**Figure supplement 2—source data 2.** Raw scan of western blots of Gαi.

observed no significant defects in mitotic spindle orientation at the ventricular surface. Additionally, no ectopic expression of neural crest markers or Wnt pathway activation was observed (***Figure 1— figure supplement 7***). Altogether, these results further indicate that non-neural cell deficiency is responsible for ectopia formation in *Ric8a* mutants.

## Microglial *Ric8a* deficiency is responsible for ectopia formation

To assess the role of non-neural cell types, we turned our attention to microglia since RNA-seq studies show that brain microglia express *Emx1* at a significant level (***Zhang et al., 2014***). To determine if *Emx1-Cre* is expressed and active in microglia, we isolated microglia from *Ric8a:Emx1-Cre* mutants. We found that *Emx1-Cre*-mediated *Ric8a* deletion resulted in altered cytokine expression in microglia (***Figure 2—figure supplement 1a, b***). This indicate that *Emx1-Cre* is expressed and active in microglia and deletes *Ric8a*. In further support of this interpretation, we found that when crossed to a reporter, *Emx1-Cre* resulted in the expression of reporter gene in microglia (***Figure 2—figure supplement 1c–c″***). It also resulted in the reduction of *Ric8a* mRNA levels in in microglia in *Ric8a:Emx1-Cre* mutants (***Figure 2—figure supplement 1d***). To determine the specific significance of *Ric8a* deletion from microglia alone, we next employed a microglia-specific *Cx3cr1-Cre* (***Yona et al., 2013***). Like *Emx1-Cre* mutants, *Ric8a:Cx3cr1-Cre* mutant microglia also showed elevated cytokine secretion and transcription in comparison to control microglia upon stimulation by lipopolysaccharide (LPS) (***Figure 2a, b***). Similar results were also obtained with stimulation by polyinosinic–polycytidylic acid (poly I:C), an

intracellular immune activator. Thus, these results indicate that *ric8a* deficiency in microglia results in broad increases in microglial sensitivity to immune stimulation.

To determine if microglial *Ric8a* deficiency alone is sufficient to cause cortical ectopia in vivo, we examined *Ric8a:Cx3cr1-Cre* mutants but found that it did not affect either basement membrane integrity or cortical layering (*Figure 2c*). We reasoned that this may be related to the fact that *Ric8a* mutant microglia only show heightened activity upon stimulation but not under basal unstimulated conditions (*Figure 2a, b*) but elevated microglial activity may be needed for basement membrane degradation and ectopia formation. To test this possibility, we employed in utero LPS administration to activate microglia during cortical development. We found that over 50% of *Ric8a:Cx3cr1-Cre* mutant neonates showed ectopia when administered LPS at E11.5–12.5 (10 of 19 mutant neonates examined) (*Figure 2d*). In contrast, no cortical ectopia were observed in any of the 32 littermate controls that were similarly administered LPS. This indicates that only the combination of microglial *Ric8a* deficiency and immune activation leads to ectopia formation. In *Emx1-Cre* mutants, ectopia develop without LPS administration (*Figure 1*). We suspect that this may be due to concurrent *Ric8a* deficiency in neural cell types, which may result in deficits that mimic immune stimulation. In the embryonic cortex, for example, studies have shown that large numbers of cells die starting as early as E12 (*Blaschke et al., 1996*; *Blaschke et al., 1998*). Radial glia and neuronal progenitors play critical roles in the clearance of apoptotic cells and cellular debris in the brain (*Amaya et al., 2015*; *Ginsty et al., 2015*; *Lu et al., 2011*) and Ric8a-dependent heterotrimeric G proteins promotes this function in both professional and non-professional phagocytic cells (*Billings et al., 2016*; *Flak et al., 2020*; *Pan et al., 2016*; *Preissler et al., 2015*; *Zhang et al., 2023*). Thus, *Ric8a* deficiency in radial glia may potentially result in accumulation of apoptotic cell debris in the embryonic brain that stimulate microglia. To test this, we next additionally deleted *Ric8a* from radial glia in the *Ric8a:Cx3cr1-Cre* microglial mutant background by introducing *Nestin-Cre*. We have shown that *Ric8a* deletion by *Nestin-Cre* alone does not result in ectopia (*Figure 1g, g'*). However, we found that, like deletion by *Emx1-Cre*, *ric8a* deletion by the dual *Cre* combination of *Cx3cr1-Cre* and *Nestin-Cre* also resulted in severe ectopia in all double *Cre* mutants (six of six examined) (*Figure 2f*). Thus, these results indicate that elevated immune activation of *Ric8a* deficient microglia during cortical development is responsible for ectopia formation.

## Microglial *APP* deficiency also results in ectopia formation

In the large numbers of cobblestone lissencephaly and related cortical ectopia mutants, besides the lateral cortex, severe ectopia are typically also observed at the cortical midline (*Beggs et al., 2003*; *Belvindrah et al., 2006*; *Graus-Porta et al., 2001*; *Huang et al., 2006*; *Moore et al., 2002*; *Niewmierzycka et al., 2005*; *Satz et al., 2010*). There are only a few mutants including the *Ric8a:Emx1-Cre* mutant that are exception, in that the ectopia are not observed at the cortical midline but are instead exclusively located to the lateral cortex (*Figure 1*). The other mutants in this unique group include the *App/Aplp1/2* triple (*Herms et al., 2004*) and *Apbb1/2* double knockouts (*Guénette et al., 2006*). This suggests that similar mechanisms involving microglia may underlie ectopia formation in *Ric8a:Emx1-Cre*, *App/Aplp1/2*, and *Apbb1/2* mutants. Independent studies also point to a role of non-neuronal cells in ectopia formation in *App* family gene mutants. For example, unlike the triple knockout, which causes neuronal over-migration, specific *App* knockdown in cortical neurons during development results in under- instead of over-migration of targeted neurons (*Young-Pearse et al., 2007*). Furthermore, Ric8a-regulated Gα proteins play a conserved role in mediating key branches of APP signaling in cells across species (*Fogel et al., 2014*; *Milosch et al., 2014*; *Nishimoto et al., 1993*; *Ramaker et al., 2013*) and we confirmed that Gαi proteins are severely depleted in *Ric8a:Emx1-Cre* mutant cortices (*Figure 2—figure supplement 2*). Thus, like in *Ric8a:Emx1-Crre* mutants, microglia may play a key role in ecotopia formation in APP pathway mutants. To test this, we first analyzed *App* mutant microglia. To this end, we employed *Cx3cr1-Cre* to delete a conditional allele of *App* from microglia and found that microglia cultured from *App:Cx3cr1-Cre* mutants showed reduced TNFα and IL-6 secretion as well as muted IL-6 transcription upon stimulation (*Figure 3a*, *Figure 3—figure supplement 1a, b*). This indicates that *App* plays a previously unrecognized, cell-autonomous role in microglia in regulating microglial activity. Microglia exhibit attenuated immune response following chronic stimulation, especially when carrying strong loss-of-function mutations in anti-inflammatory pathways (*Chamberlain et al., 2015*; *Sayed et al., 2018*). We suspect that the attenuated response by *App* mutant microglia may result from similar effects following in vitro culture. To test effects of

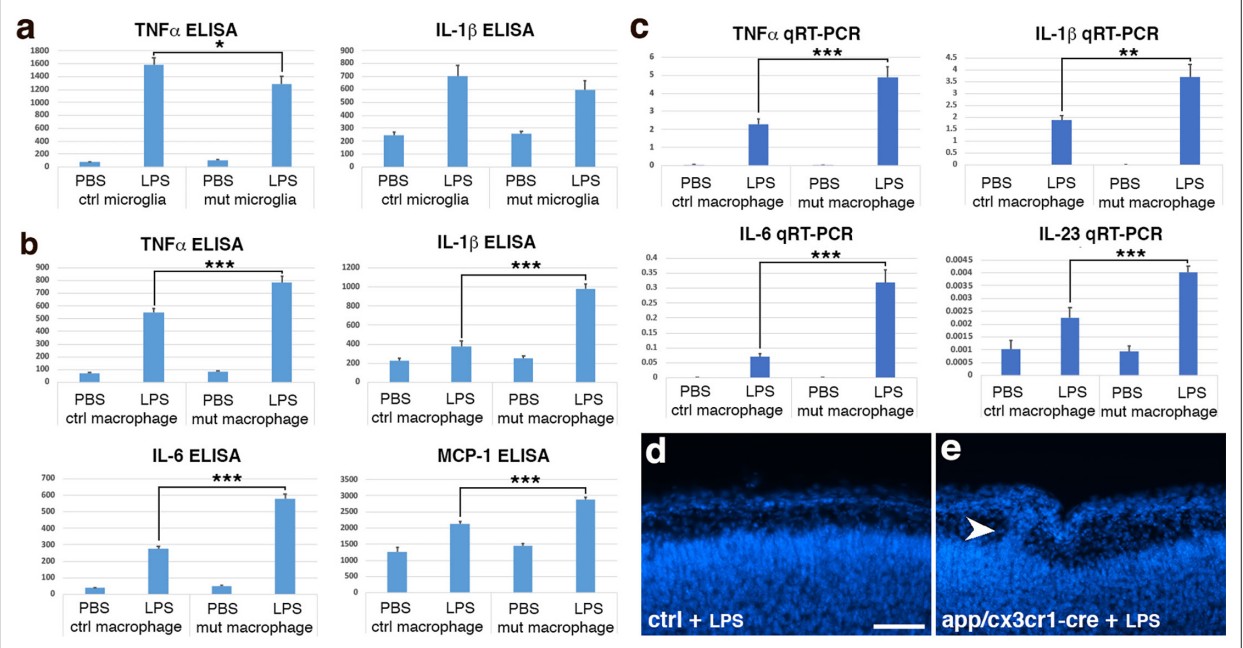

**Figure 3.** *App* deficiency results in hypersensitive microglia and cortical ectopia. (**a**) TNFα and IL-1β secretion (pg/ml) in cultured control and *App:Cx3cr1-Cre* mutant microglia following lipopolysaccharide (LPS) stimulation. *p < 0.05; n = 7–9 each group. (**b**) TNFα, IL-1β, IL-6, and MCP1 secretion (pg/ml) in fresh unelicited control and *App:Cx3cr1-Cre* mutant peritoneal macrophages following LPS stimulation. ***p < 0.001; n = 7–10 each group. (**c**) TNFα, IL-1β, IL-6, and IL-23 mRNA expression in fresh unelicited control and *App:Cx3cr1-Cre* mutant peritoneal macrophages following LPS stimulation. **p < 0.01; ***p < 0.001; n = 6 each group. Nuclear (DAPI, in blue) staining of control (**d**) and LPS-treated *App:Cx3cr1-Cre* mutant (**e**) cortices at P0. Note cortical ectopia in the mutant cortex (arrowhead). Scale bar in (**d**), 200 μm for (**d, e**).

The online version of this article includes the following source data and figure supplement(s) for figure 3:

**Source data 1.** Excel files for control and App mutant microglia/macrophage ELISA and qRT-PCR analysis.

**Figure supplement 1.** Cytokine secretion and transcriptional induction in *App:Cx3cr1-Cre* mutant microglia.

*App* mutation under conditions that more closely resemble in vivo physiological conditions, we next isolated fresh, unelicited peritoneal macrophages and acutely analyzed their response to immune stimulation. We found that *App* mutant macrophages showed significantly elevated secretion of all cytokines tested (*Figure 3b*). At the transcriptional level, mRNA induction was also increased for all cytokines (*Figure 3c*). Thus, like that of *Ric8a*, the normal function of *App* also appears to be to suppress the inflammatory activation of microglia.

To determine if microglial *app* deficiency is also responsible for ectopia formation in *app* triple knockout mutants, we next asked if activating microglia in microglia-specific *App* mutants similarly results in pial ectopia during cortical development. To this end, we administered LPS in utero at E11.5–12.5 to *App:Cx3cr1-Cre* mutant animals as we did to *Ric8a:Cx3cr1-Cre* mutants above. We found that, while none of the 81 littermate controls administered LPS showed ectopia, a significant number of mutant neonates showed ectopia (6 of 31 neonates examined, ~19%) and associated breaches in the basement membrane (*Figure 3e*, *Figure 3—figure supplement 1c–f"*). Thus, *app* deficient microglia, when activated, also results in cortical ectopia during development. The reduced severity of the ectopia observed, as compared to that in *Ric8a:Cx3cr1-Cre* mutants, likely in part results from the reduced LPS dosage (by ~threefolds) we had to use in these animas due to the enhanced immune sensitivity of their strain genetic background. Other *App* gene family members are also expressed in microglia (*Zhang et al., 2014*) and ectopia are only observed in *App/Aplp1/2* triple but not in any double or single mutants (*Herms et al., 2004*). Aplp1/2 may therefore also compensate for the loss of APP in microglia. Thus, these results indicate that *App* normally plays a cell-autonomous role in microglia that negatively regulate microglial activation, and its loss of function underlies cortical ectopia formation. The similarities between *App* and *Ric8a* mutant phenotypes suggest that they form a previously unknown anti-inflammatory pathway in microglia.

# Monomeric Aβ suppresses microglial inflammatory activation via an APP–Ric8a pathway

The possibility that *App* and *Ric8a* may form a novel anti-inflammatory pathway in microglia raises questions on the identity of the ligands for the pathway. Several molecules have been reported to bind to APP and/or activate APP-dependent pathways (*Fogel et al., 2014*; *Milosch et al., 2014*; *Rice et al., 2012*), among which Aβ is noteworthy for its nanomolar direct binding affinity (*Fogel et al., 2014*; *Shaked et al., 2006*). Aβ oligomers and fibrils have been shown by numerous studies to be pro-inflammatory, while non-fibrillar Aβ lack such activity (*Halle et al., 2008*; *Huang, 2023*; *Huang, 2024*; *Lorton et al., 1996*; *Muehlhauser et al., 2001*; *Tan et al., 1999*). In contrast, when employed under conditions that favor the monomer conformation, Aβ inhibits T cell activation (*Grant et al., 2012*). This suggests that, unlike pro-inflammatory Aβ oligomers (*Figure 4—figure supplement 1j*), Aβ monomers may be anti-inflammatory. To test this possibility, we dissolved Aβ40 peptides in dimethyl sulfoxide (DMSO), a standard approach in Alzheimer's disease research that has been shown to preserve the monomeric conformation (*LeVine, 2004*; *Stine et al., 2011*). We found that Aβ monomers as prepared potently suppressed the secretion of large numbers of cytokines (*Figure 4a*, *Figure 4—figure supplement 1*) and showed similar effects on microglia no matter if they were activated by LPS or poly I:C (*Figure 4b*). We also found that the Aβ monomers similarly strongly inhibited the induction of cytokines at the transcriptional level (*Figure 4c*, *Figure 4—figure supplement 1*). In addition, we observed these effects with Aβ40 peptides from different commercial sources. Thus, these results indicate that monomeric Aβ possesses a previously unreported anti-inflammatory activity against microglia that strongly inhibits microglial inflammatory activation.

To determine whether monomeric Aβ signals through APP, we employed *App:Cx3cr1-Cre* mutant microglia. We found that, unlike that of control microglia, Aβ monomers failed to suppress the secretion of all tested cytokines by *App* mutant microglia (*Figure 4d*, *Figure 4—figure supplement 1*). Interestingly, this blockade appeared to be specific to *App* since Aβ monomers still significantly suppressed cytokine secretion by *Aplp2* mutant microglia. At the transcriptional level, Aβ monomers also failed to suppress cytokine induction in *App* mutant microglia (*Figure 4e*, *Figure 4—figure supplement 1*). Together, these results indicate that APP is functionally required in microglia for Aβ monomer inhibition of cytokine expression at both transcriptional and post-transcriptional levels. Cultured microglia from *App:Cx3cr1-Cre* mutants showed attenuated immune activation (*Figure 3*). To assess whether this may affect the efficacy of Aβ monomer inhibition, we next tested the response of fresh, unelicited macrophages. We found that, like that of control microglia, cytokine secretion by control macrophages was also strongly suppressed by Aβ monomers (*Figure 4f*, *Figure 4—figure supplement 1*). However, even though *App* mutant macrophages showed elevated response to immune stimulation in comparison to control macrophages, they still failed to respond to Aβ monomers and displayed levels of cytokine secretion that were indistinguishable from those of DMSO-treated cells (*Figure 4f*, *Figure 4—figure supplement 1*). Thus, these results further indicate that APP function is required in microglia for mediating the anti-inflammatory effects of Aβ monomers.

The similarity of *Ric8a* ectopia to *App* ectopia phenotype (*Figures 2 and 3*) also suggests that Ric8a functions in the same pathway as APP in mediating Aβ monomer anti-inflammatory signaling in microglia. This is consistent with previous studies showing that heterotrimeric G proteins are coupled to APP and mediate APP intracellular signaling in vitro and vivo (*Fogel et al., 2014*; *Milosch et al., 2014*; *Nishimoto et al., 1993*; *Ramaker et al., 2013*) and that Ric8a is a molecular chaperone essential for the post-translational stability of heterotrimeric G proteins (*Gabay et al., 2011*; *Tall et al., 2003*). To directly test if Ric8a is part of this pathway, we next employed *Ric8a:Cx3cr1-Cre* mutant microglia. We found that, indeed, like that of *App* mutant microglia, Aβ monomers also failed to suppress the secretion of TNFα and IL-6 by *Ric8a* mutant microglia (*Figure 4g*). This indicates that heterotrimeric G proteins function is likely required in the same pathway of APP in microglia for the suppression of TNFα and IL-6 secretion. However, unlike APP, we found that Ric8a appears to be dispensable for Aβ monomer regulation of other cytokines. For example, unlike that of TNFα and IL-6, Aβ monomers still suppressed IL-1β secretion by *Ric8a* mutant microglia (*Figure 4—figure supplement 1*). It also appears to be dispensable for the regulation of cytokine transcription since Aβ monomers similarly suppressed IL-6 transcriptional induction in both control and *Ric8a* mutant microglia. These results suggest that heterotrimeric G proteins function may only

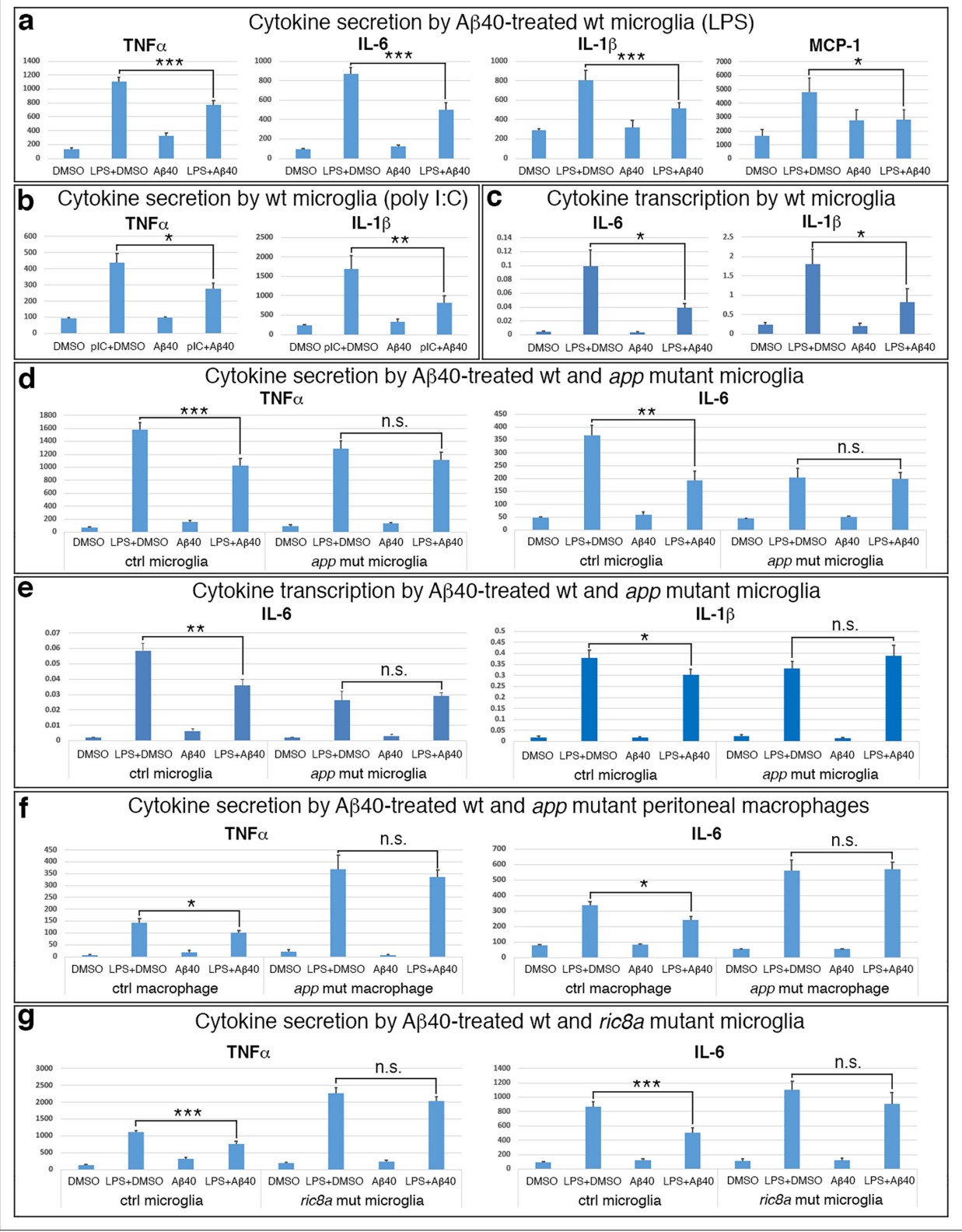

**Figure 4.** Monomeric Aβ40 suppresses microglia via APP and Ric8a. (**a**) TNFα, IL-6, IL-1β, and MCP1 secretion (pg/ml) by wildtype microglia following lipopolysaccharide (LPS) stimulation in the absence or presence of Aβ40 (200 or 500 nM). *p < 0.05; ***p < 0.001; n = 8–14 each group. (**b**) TNFα and IL-1β secretion (pg/ml) by wildtype microglia following poly I:C stimulation in the absence or presence of Aβ40 (500 nM). *p < 0.05; **p < 0.01; n = 6–7 each group. (**c**) IL-6 and IL-1β mRNA induction in wildtype microglia following LPS stimulation in the absence or presence of Aβ40 (500 nM). *p <

*Figure 4 continued on next page*

*Figure 4 continued*

0.05; *n* = 6 each group. (**d**) TNFα and IL-6 secretion (pg/ml) by control and *App:Cx3cr1-Cre* mutant microglia following LPS stimulation in the absence or presence of Aβ40 (200 nM). **p < 0.01; ***p < 0.001; *n* = 8 each group. (**e**) IL-6 and IL-1β mRNA induction in control and *App:Cx3cr1-Cre* mutant microglia following LPS stimulation in the absence or presence of Aβ40 (200 nM). *p < 0.05; **p < 0.01; *n* = 6 each group. (**f**) TNFα and IL-6 secretion (pg/ml) by control and *App:Cx3cr1-Cre* mutant peritoneal macrophages following LPS stimulation in the absence or presence of Aβ40 (500 nM). *p < 0.05; *n* = 6–7 each group. (**g**) TNFα and IL-6 secretion (pg/ml) by control and *Ric8a:Cx3cr1-Cre* mutant microglia following LPS stimulation in the absence or presence of Aβ40 (200 nM). ***p < 0.001; *n* = 12–14 each group.

The online version of this article includes the following source data and figure supplement(s) for figure 4:

**Source data 1.** Excel files for control and App and Ric8a mutant microglia/macrophage ELISA and qRT-PCR analysis undergoing Aβ40 stimulation.

**Figure supplement 1.** Effects of monomeric amyloid β (Aβ) on cytokine secretion and transcription in control and mutant microglial lineage cells.

mediate some of the anti-inflammatory signaling of monomeric Aβ. Thus, APP and Ric8a-regulated heterotrimeric G proteins form part of a novel anti-inflammatory pathway activated by monomeric Aβ in microglia.

## Elevated matrix metalloproteinases cause basement membrane degradation

We have shown that heightened microglial activation due to mutation in the Aβ monomer-activated APP/Ric8a pathway results in basement membrane degradation and ectopia during cortical development. To further test this interpretation, we sought to test the prediction that inhibition of microglial activation in these mutants suppressed the formation ectopia. To this end, we employed dorsomorphin and S3I-201, inhibitors targeting Akt, Stat3, and other mediators in pro-inflammatory signaling (*Lee et al., 2016*; *Qin et al., 2012*). Consistent with their anti-inflammatory activity, we found that dorsomorphin and S3I-201 both suppressed astrogliosis associated with neuroinflammation in the cortex of *Ric8a:Emx1-Cre* mutants (*Figure 5—figure supplement 1*). Furthermore, they also suppressed the formation of ectopia in *Ric8a:Emx1-Cre* mutants, reducing both the number and the size of the ectopia observed (*Figure 5a–f*, *Figure 5—figure supplement 2*). Most strikingly, the combined administration of dorsomorphin and S3I-201 nearly eliminated all ectopia in *Ric8a:Emx1-Cre* mutants (*Figure 5d, e*). Thus, these results indicate that excessive inflammatory activation of microglia is responsible for ectopia formation in *ric8a* mutants.

Under neuroinflammatory conditions, brain cytokines frequently induce matrix metalloproteinases (MMPs), which lead to breakdown of the extracellular matrix and contribute to disease pathology (*Pagenstecher et al., 1998*; *Wang et al., 2000*). Since *Ric8a* mutant microglia are hyperactive in inflammatory cytokine production, we wonder if induction of MMPs may underlie the laminin degradation and cortical basement membrane break observed in *Ric8a* mutants. To test this, we examined MMP9 expression in the embryonic cortex by in situ hybridization. We found that at E13.5, MMP9 mRNA is strongly expressed in a sparse cell population resembling microglia populating the cortex at this stage (*Squarzoni et al., 2014*; *Figure 5—figure supplement 2*). Next, we examined the activities of MMP2 and MMP9 in the developing control and mutant cortex using gelatin gel zymography. We found that the activity of MMP9 in the mutant cortex was significantly increased (*Figure 5i*, *Figure 5—figure supplement 2*). In contrast, that activity of MMP2 remained unaffected. Similarly, at the protein level, we found that the immunoreactivity for MMP9 was increased in *Ric8a:Emx1-Cre* mutants (*Figure 5g, h*). To test if the increased MMP activity is responsible for the ectopia observed, we next employed BB94, a broad-spectrum inhibitor of MMPs. We found that BB94 administration significantly suppress both the number and the size of the ectopia in *ric8a* mutants (*Figure 5j–m*). To narrow down the identity of MMPs responsible, we further employed an inhibitor specific for MMP9 and 13 (MMP-9/MMP-13 inhibitor I, CAS 204140-01-2) and found that it similarly suppressed both the number and the size of the ectopia (*Figure 5l, m*). Furthermore, consistent with its near complete suppression of cortical ectopia (*Figure 5a–f*), we found that the co-administration of dorsomorphin and S3I-201 also reduced MMP9 activity in the mutant cortex to the control level (*Figure 5—figure supplement 3*). Thus, these results indicate this Aβ monomer-regulated anti-inflammatory pathway normally promotes cortical development through suppressing microglial activation and MMP induction.

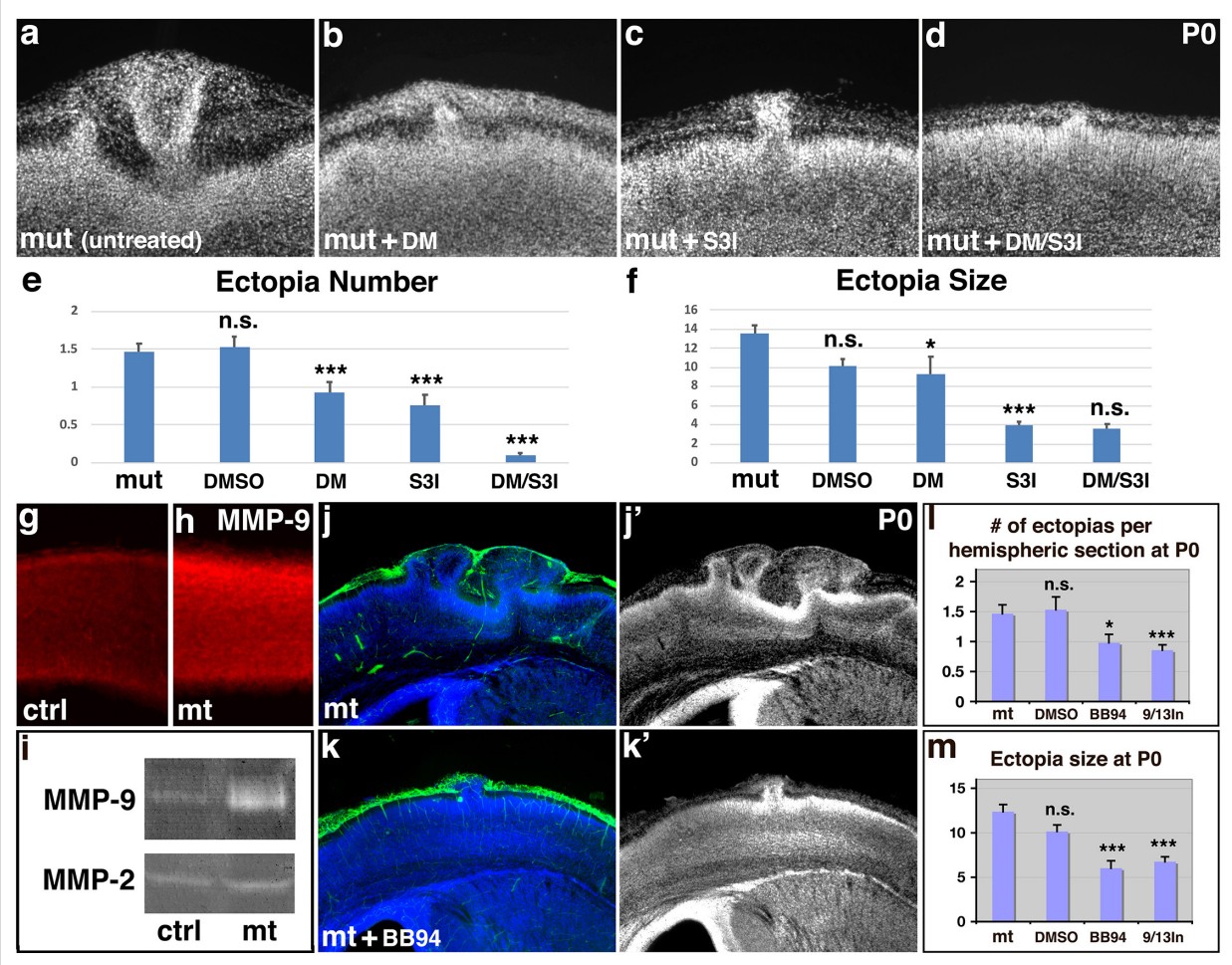

**Figure 5.** Inhibition of both microglial inflammatory activation and cortical MMP9 activity suppresses basement membrane breach and neuronal ectopia. Nuclear (DAPI, in gray) staining of untreated (**a**), anti-inflammatory drug dorsomorphin (DM) (**b**), Stat3 inhibitor S3I-201 (S3I) (**c**), and DM/S3I (**d**) dual treated *Ric8a:Emx1-Cre* mutant cortices at P0. Quantitative analysis of ectopia number (**e**) and size (**f**) in the neonatal mutant cortex after DMSO, DM, S3I, and DM/S3I dual treatment at E12.5. *p < 0.05; ***p < 0.001; all compared to untreated mutants. The reduction in ectopia size after dual treatment is not statistically significant, likely due to the small number of ectopias that remained. MMP9 (in red) staining of control (**g**) and mutant cortices (**h**) at E13.5. Quantification shows statistically significant increases in mutants (control, 24.8 ± 0.2 AU [arbitrary units]; mutant, 35.7 ± 1.7 AU; p = 0.002; *n* = 6). (**i**) Gel zymography of control and mutant cortical lysates at E13.5. Increased levels of MMP9 but not of MMP2 were observed in mutants (control, 1.00 ± 0.06 AU; mutant, 3.72 ± 1.86 AU; p = 0.028; *n* = 4). See further details in *Figure 5—figure supplements 2 and 3*. (**j–k'**) Laminin (in green) and nuclear (DAPI, in blue) staining of mutant cortices untreated (**h**) or treated (**I**) with BB94. (**l, m**) Quantitative analysis of ectopia number and size following MMP inhibitor BB94 or MMP9/13 inhibitor I treatment. *p < 0.05; ***p < 0.001; all compared to untreated mutants.

The online version of this article includes the following source data and figure supplement(s) for figure 5:

**Source data 1.** Excel files for Ric8a:Emx1-cre mutant ectopia suppression analysis.

**Figure supplement 1.** Suppression of astrogliosis in *Ric8a:Emx1-Cre* mutant cortices by anti-inflammatory drugs, dorsomorphin (DM) and S3I-201 (S3I).

**Figure supplement 2.** MMP9 in situ and activity in E13.5 *Ric8a:Emx1-Cre* mutant cortices.

**Figure supplement 2—source data 1.** Whole gel gelatin zymography images of Ric8a:Emx1-Cre control and mutant cortices.

**Figure supplement 2—source data 2.** Raw scan of zymography data.

**Figure supplement 3.** Suppression of MMP9 expression in *Ric8a:Emx1-Cre* mutant cortices by anti-inflammatory drugs, dorsomorphin (DM) and S3I-201 (S3I).

## Discussion

The spatiotemporal expression of immune cytokines by glial cells in the brain plays critical roles in the normal development, function, and plasticity of the brain circuitry (*Barres, 2008*; *Schafer and Stevens, 2015*; *Zipp et al., 2023*). In this article, we have identified a novel microglial anti-inflammatory

pathway activated by monomeric Aβ that inhibits microglial cytokine expression and plays essential roles in the normal development of the cerebral cortex. We have found that this pathway is mediated by APP and the heterotrimeric G protein GEF and molecular chaperone Ric8a in microglia and its activation leads to the inhibition of microglial cytokine induction at transcriptional and post-transcriptional levels (*Figures 1–4*). We further show that a key function of this pathway is to suppress the activity of MMP9 during corticogenesis and disruption of this regulation results in cortical basement membrane degradation and neuronal ectopia development (*Figures 1–3 and 5*). Furthermore, we find that this pathway is activated specifically by the monomeric form of Aβ in vitro (*Figure 4*), identifying, for the first time, an isoform-specific activity of Aβ against microglia. These results provide novel insights into the neuron–glia communication mechanisms that coordinate the regulation of immune cytokines, key regulators of Hebbian and non-Hebbian synaptic plasticity, by glial cells in the brain. The discovery of the novel activity of monomeric Aβ as a negative regulator of microglia may also facilitate the further elucidation of Alzheimer's disease pathogenesis.

## Microglial activity regulation during cortical development

Among the glial cell populations in the brain, astrocytes and oligodendrocyte are both born within the nervous system at the end of cortical neurogenesis. As such, they play limited roles in the early steps of cortical development. In contrast, microglia are not only of a distinct non-neural lineage that originates from outside the nervous system but also begin to populate the brain at the onset of corticogenesis (*Ginhoux et al., 2010*; *Hattori et al., 2023*). As such, they play unique roles throughout cortical development. Indeed, microglial activity has been found to regulate the size of the cortical neural precursor pool (*Cunningham et al., 2013*). Microglia-secreted cytokines have also been found to promote both neurogenesis and oligodendrogenesis (*Shigemoto-Mogami et al., 2014*). As such, the precise regulation of microglial activity is critical to the normal development of the neocortex from an early stage. In this study, we have shown that immune over-activation of microglia deficient in a monomeric Aβ-regulated pathway results in excessive cortical matrix proteinase activation, leading basement membrane degradation and neuronal ectopia. Previous studies have shown that reductions in the expression of microglial immune and chemotaxis genes instead lead to the failure of microglia to populate the brain (*Iyer et al., 2022*). These results together highlight the importance of precisely regulating the level of microglial activity during brain development. The dramatic destructive effects of microglial hyperactivity observed during corticogenesis also foreshadow the critical roles it plays in brain dysfunction and disease at later stages of life.

In this study, we have also shown that the anti-inflammatory regulation of microglia in corticogenesis depends on a pathway composed of APP and the heterotrimeric G protein regulator Ric8a. This has revealed new insight into the intercellular signaling mechanisms regulating microglial activity in the brain. Heterotrimeric G proteins are well-known mediators of G-protein-coupled receptor signaling. In this study, we have found that they likely also function in the same pathway as APP. To our knowledge, ours is the first study to report an in vivo anti-inflammatory function of this pathway in microglia and has significantly advanced knowledge in microglial biology. This is also consistent with previous studies showing that heterotrimeric G proteins directly interact with the APP cytoplasmic domain and mediate key branches of APP signaling from invertebrates to mammals (*Fogel et al., 2014*; *Milosch et al., 2014*; *Nishimoto et al., 1993*; *Ramaker et al., 2013*). In this study, we have in addition shown that this pathway is specifically activated in vitro by the monomeric form of Aβ, a peptide produced by neurons in the brain (*Cirrito et al., 2005*), providing further insight into the biological function of this pathway. In the early cortex, neurogenesis is just beginning, and most neurons born are in an immature state. It is unclear if this pathway is activated by Aβ at this stage in vivo. However, studies have shown that other APP ligands such as pancortin, a member of the olfactomedin family proteins known to inhibit innate immunity (*Liu et al., 2010*), are expressed in the cortex at this stage (*Rice et al., 2012*). It will be interesting to determine if these innate immune regulators play a role in regulating this pathway.

In this and previous studies, we have found that deletion of *Ric8a* gene from radial glial progenitors using *Nestin-Cre* does not result in obvious cortical ectopia (*Figure 2*; *Ma et al., 2017*). However, when *Ric8a* is in addition deleted from microglia, this results in severe cortical ectopia (*Figure 2*), implicating a novel role of microglia in cortical ectopia development. Previous studies have reported that *Ric8a* deletion by *Nestin-Cre* alone results in cortical ectopia (*Kask et al., 2015*; *Kask et al., 2018*). The

cause for this discrepancy is at present unclear. The expression of *Nestin-Cre*, however, is known to be influenced by several factors including transgene insertion site and genetic background and the same *Nestin-Cre* has been reported to be active and induce gene inactivation in microglia (*Karasinska et al., 2013*; *Takamori et al., 2009*). These factors may play a role in this discrepancy. In our studies, we show that microglia-specific *Ric8a* deletion using *Cx3cr1-Cre* during development results in severe cortical ectopia upon and only upon immune stimulation (*Figure 2*). We further show that microglia-specific *App* deletion results in similar ectopia also only upon immune stimulation (*Figure 3*). These results are important findings as they implicate, for the first time, a causative role played by microglial dysfunction in the formation of cortical ectopia in neurodevelopmental disorders.

## Neuronal activity, glial cytokine expression, and brain circuit plasticity

Activity-dependent competitive and homeostatic plasticity is a foundational rule that regulates the development, maturation, and function of neural circuits across brain regions. Studies have shown that glial cells, through regulating the spatiotemporal expression of immune cytokines, play a pivotal role in this process. In the developing thalamus, by activating interleukin-33 expression in an activity-dependent manner, astrocytes have been found to promote the segregation of eye-specific axonal projection and the maturation of the visual circuitry (*He et al., 2022*; *Vainchtein et al., 2018*). In the visual cortex, astrocytic expression of TNFα similarly mediates activity-dependent homeostatic upscaling of cortical synapses following peripheral monocular deprivation (*Barnes et al., 2017*; *Heir et al., 2024*; *Kaneko et al., 2008*). In this study, we have shown that Aβ monomers inhibit expression of cytokines by brain microglia via a novel APP/heterotrimeric G-protein-mediated pathway. Aβ is primarily produced by neurons in the brain in a neural activity-dependent manner and form oligomers when large quantities are produced (*Cirrito et al., 2005*). Aβ oligomers, in contrary to monomers, are pro-inflammatory and increase glial cytokine expression (*Halle et al., 2008*; *Huang, 2023*; *Lorton et al., 1996*; *Muehlhauser et al., 2001*; *Tan et al., 1999*). These findings thus suggest that different levels of neural circuit activity in the brain may differentially regulate glial cytokine expression through inducing different levels of Aβ. High levels of neural activity may lead to high levels of Aβ and the formation of Aβ oligomers that activate glial cytokine production, while low levels of neural activity may produce low levels of Aβ, maintain Aβ as monomers, and inhibit glial cytokine production. Thus, Aβ in the brain may not only be a reporter of the levels of neural circuit activity but may also serve as an agent that directly mediate activity level-dependent plasticity. Following sensory deprivation, for example, Aβ levels may be lowered due to loss of sensory stimulation. This may lead to the relief of monomeric Aβ inhibition of cytokines such as TNFα and as a result trigger homeostatic upscaling of cortical synapses in the visual cortex (*Barnes et al., 2017*; *Heir et al., 2024*; *Kaneko et al., 2008*). In contrary, when neural activity levels are high, large quantities of Aβ may be produced, leading to formation of Aβ oligomers that may in turn induce expression of cytokines such as IL-33 that promote synaptic pruning. A large body of evidence strongly indicates that Aβ and related pathways indeed mediate homeostatic and competitive plasticity in the visual and other systems of the brain (*Galanis et al., 2021*; *Huang, 2023*; *Huang, 2024*; *Kamenetz et al., 2003*; *Kim et al., 2013*). Our discovery of the Aβ monomer-activated pathway has therefore provided novel insights into a universal mechanism that senses neural circuit activity pattern and translates it into homeostatic and competitive synaptic changes in the brain, a mechanism with fundamental roles in cognitive function.

In this study, we have also found that the matrix proteinase MMP9 is a key downstream effector of microglial activity in the developing cortex. We find that microglial hyperactivity results in increased levels of MMP9, leading to cortical basement membrane degradation and neuronal ectopia and inhibiting MMP9 directly or indirectly suppresses the phenotype. This suggests that the regulation of MMP9 may be a key mechanism by which glial cells regulate brain development and plasticity. Indeed, independent studies have shown that, in the visual cortex, MMP9 is also a pivotal mediator of TNFα-dependent homeostatic upscaling of central synapses following monocular deprivation (*Akol et al., 2022*; *Kaneko et al., 2008*; *Kelly et al., 2015*; *Spolidoro et al., 2012*). In the *Xenopus* tectum, MMP9 has similarly been found to be induced by neural activity and promote visual activity-induced dendritic growth (*Gore et al., 2021*). Importantly, in both wildtype and amblyopic animals, light reintroduction after dark exposure has been found to reactivate plasticity in the adult visual cortex via MMP9, uncovering a potential treatment for common visual conditions (*Murase et al., 2017*; *Murase et al., 2019*). These results therefore highlight a conserved glia/cytokine/MMP9-mediated mechanism

that regulates brain development and plasticity from embryogenesis to adulthood. In ocular dominance plasticity, MMP9 is activated at perisynaptic regions (*Murase et al., 2017*; *Murase et al., 2019*). MMP9 mRNA translation has been also observed in dendrites (*Dziembowska et al., 2012*). In the *Ric8a* mutant cortex, we find that MMP9 activity is increased. Further studies are required to precisely determine the cellular sources of MMP9 and how its activity is regulated.

## Aβ monomer anti-inflammatory activity and Alzheimer's disease

Aβ is well known as a component of the amyloid plaques in the Alzheimer's disease brain. It is a unique amphipathic peptide that can, dependent on concentration and other conditions, remain as monomers or form oligomers. Studies on Aβ have historically focused on the neurotoxic effects of Aβ oligomers and their pro-inflammatory effects on glia (*Gulisano et al., 2018*; *Halle et al., 2008*; *He et al., 2019*; *Huang, 2023*; *Kim et al., 2013*; *Laurén et al., 2009*; *Lazarevic et al., 2017*; *Lorton et al., 1996*; *Muehlhauser et al., 2001*; *Parodi et al., 2010*; *Puzzo et al., 2008*; *Shankar et al., 2008*; *Tan et al., 1999*; *Walsh et al., 2002*; *Yang et al., 2015*; *Zott et al., 2019*). In this study, we have found that, in contrary to Aβ oligomers, Aβ monomers instead possess a previously unknown anti-inflammatory activity that acts through a unique microglial pathway. We have further found that genetic disruption of this pathway in corticogenesis results microglial hyperactivity, leading to neuronal ectopia and large disruption of cortical structural organization. To our knowledge, ours is the first study to uncover this overlooked anti-inflammatory activity of Aβ monomers. It is in alignment with recent studies showing that Aβ monomers are also directly protective to neurons and positively regulate synapse development and function (*Galanis et al., 2021*; *Giuffrida et al., 2009*; *Plant et al., 2003*; *Ramsden et al., 2002*; *Zhou et al., 2022*). Assuming a set amount of Aβ peptides, the formation of Aβ oligomers and aggregates in the brain would, by chemical law, be predicted to result in the depletion of Aβ monomers (*Dear et al., 2020*; *Michaels et al., 2020*). Thus, in the Alzheimer's disease brain, besides the obvious formation of Aβ aggregates, there may also be a less visible depletion of Aβ monomers taking place at the same time, which may, like Aβ oligomers, also contribute to the development of neuroinflammation and neuronal damage (*Huang, 2023*). In support of this interpretation, high soluble brain Aβ42, which likely also means high levels of Aβ monomers in the brain, have been found in clinical studies to preserve cognition in patients of both familial and sporadic Alzheimer's disease, in spite of increasing amyloidosis detected in their brains (*Espay et al., 2021*; *Sturchio et al., 2022*; *Sturchio et al., 2021*). In our study, we have also found that the effects of microglial disinhibition are mediated by MMP9. Importantly, in neurodegenerative diseases, MMP9 has been similarly found to be a key determinant regulating the selective degeneration of neuronal cell types (*Kaplan et al., 2014*; *Tran et al., 2019*). MMP9 levels are also upregulated in the plasma in both mild cognitive impairment and Alzheimer's disease patients (*Bruno et al., 2009*; *Lorenzl et al., 2008*; *Tsiknia et al., 2022*). In addition, in several motor neuron disease models, reducing MMP9 has been found to protect neurons and delay the loss of motor function (*Kaplan et al., 2014*; *Spiller et al., 2019*). Thus, our study has not only uncovered a potentially overlooked role of Aβ monomer depletion in the development of Alzheimer's disease but also identified downstream effectors. Elucidating the roles these factors play may reveal new insight into the pathogenesis of Alzheimer's disease.

## Methods

### Generation of *Ric8a* conditional allele

Standard molecular biology techniques were employed for generating the conditional *Ric8a* allele. Briefly, genomic fragments, of 4.5 and 2.5 kb and flanking exons 2–4 of the *Ric8a* locus at the 5′ and 3′ side, respectively, were isolated by PCR using high fidelity polymerases. Targeting plasmid was constructed by flanking the genomic fragment containing exons 2–4 with two loxP sites together with a *neomycin*-positive selection cassette, followed by 5′ and 3′ genomic fragments as homologous recombination arms and a *pgk-DTA* gene as a negative selection cassette. ES cell clones were screened by Southern blot analysis using external probes at 5′ and 3′ sides. For derivation of conditional allele, the *neomycin* cassette was removed by crossing to an *Actin-Flpe* transgenic line after blastocyst injection and germ line transmission. The primer set for genotyping *ric-8a* conditional allele, which produces a wildtype band of ~110 bp and a mutant band of ~200 bp, is: 5′-cctagttgtgaatcag

aagcacttg-3′ and 5′-gccatacctgagttacctaggc-3′. Animals homozygous for the conditional *ric-8a* allele are viable and fertile, without obvious phenotypes.

## Mouse breeding and pharmacology

*Emx1-Cre* (IMSR_JAX:005628), *Nestin-Cre* (IMSR_JAX:003771), *Foxg1-Cre* (IMSR_JAX:004337), *Cx3cr1-Cre* (IMSR_JAX:025524), floxed *App* (IMSR_JAX:030770) as well as the *BAT-lacZ* (IMSR_JAX:005317) reporter mouse lines were purchased from the Jackson Lab. *Nex-Cre* and *Wnt3a-Cre* were as published (*Goebbels et al., 2006*; *Yoshida et al., 2006*). *Cre* transgenes were introduced into the *Ric8a or App* conditional mutant background for phenotypic analyses and *Ric8a or App* homozygotes without *Cre* as well as heterozygotes with *Cre* (littermates) were both analyzed as controls. For BB94 and MMP9/13 inhibitor injection, pregnant females were treated daily from E12.5 to E14.5 at 30 µg (BB94) or 37.5 µg (MMP9/13 inhibitor) per g of body weight. For dorsomorphin and S3I-201 injection, pregnant females were treated on E12.5 at 7.5 and 25 µg per g of body weight, respectively. For sham treatment, pregnant females were treated on E12.5 with 100 µl of DMSO. BrdU was injected at 100 µg per g of body weight, and embryos were collected 4 hr later for cell proliferation analysis, or alternatively, pups were sacrificed at P5 for neuronal migration analysis and at P17 for other analysis. For LPS treatment, pregnant females were injected intraperitoneally with 400 ng (*Ric8a* genetic background) or 150 ng (*App* genetic background) LPS per g of body weight on both E11.5 and E12.5. Animal use was in accordance with institutional guidelines.

## Immunohistochemistry

Vibratome sections from brains fixed in 4% paraformaldehyde were used. The following primary antibodies were used at respective dilutions/concentrations: mouse anti-BrdU supernatant (clone G3G4, Developmental Studies Hybridoma Bank [DSHB], University of Iowa, IA; 1:40), mouse anti-RC2 supernatant (DSHB; 1:10), mouse anti-Nestin supernatant (DSHB; 1:20), mouse anti-Vimentin supernatant (DSHB; 1:10), mouse anti-Pax6 supernatant (DSHB; 1:20), moue anti-Reelin (Millipore, 1:500), mouse anti-chondroitin sulfate (CS-56, Sigma, 1:100), rat anti-Ctip2 (Abcam, 1:500), rabbit anti-phospho Histone H3 (Ser10) (Millipore; 1:400), rabbit anti-Cux1 (CDP) (Santa Cruz; 1:100), rabbit anti-laminin (Sigma; 1:2000), rabbit anti-GFAP (Dako;1:1000), rabbit anti-ALDH1L1 (Abcam, 1:500), rabbit anti-MMP9 (Abcam, 1:1000), goat anti-MMP2 (R&D Systems; 5 µg/ml), rabbit anti-Calretinin (Chemicon, 1:2000), mouse anti-S100β (Thermo Scientific; 1:100), rabbit anti-S100β (Thermo Scientific; 1:200), and rabbit anti-phospho-Smad1/5 (Ser463/465) (41D10; Cell Signaling, 1:200). FITC- and Cy3-conjugated secondary antibodies were purchased from Jackson ImmunoResearch Laboratories (West Grove, PA). Peroxidase-conjugated secondary antibodies were purchased from Santa Cruz Biotech. Staining procedures were performed as described previously (*Huang et al., 2006*), except for anti-Ric-8a, MMP9, and phospho-Smad1/5 staining, in which a tyramide signal amplification plus Cy3 kit (PerkinElmer, Waltham, MA) was used per the manufacturer's instruction. Sections were mounted with Fluoromount G medium (Southern Biotech, Birmingham, AB) and analyzed under a Nikon *eclipse* Ti microscope or an Olympus confocal microscope.

## Microglia culture and assay

Cerebral hemispheres were dissected from individual neonates, mechanically dissociated, split into three to four wells each and cultured in DMEM-F12 (Lonza) containing 10% fetal bovine serum (Invitrogen). Microglial cells were harvested by light trypsinization that removes astroglial sheet on days 13–15. For experiments other than assaying IL-1β secretion, microglia were treated with LPS at 20 ng/ml for 3 hr or at 5 ng/ml overnight and, if applicable, DMSO or Aβ40 (ApexBio and Genscript) was applied at the same time as LPS. For assaying IL-1β secretion, microglia were primed with LPS at 200 ng/ml for 5–6 hr before treatment with 3 mM ATP for 15 min. In these experiments, DMSO or Aβ40 was applied at the same time as ATP if applicable. Supernatants were collected and used for cytokine ELISA assays per manufacturer's instructions (Biolegend). Total RNAs were prepared from collected cells using Trizol (Invitrogen) and cDNAs were synthesized using a High-capacity cDNA reverse transcription kit (Applied Biosystems). Quantitative PCR was performed using a GoTaq qPCR master mix per manufacturer's instructions (Promega). All gene expression levels were normalized against that of GAPDH.

## Quantitative and statistical analysis

The sample size was estimated to be 3–9 animals each genotype (every fourth of 50 μm coronal sections, 7–10 sections each animal) for ectopia analysis, 3–5 animals each genotype (3–4 sections each animal) for immunohistochemical analysis, and 4–6 animals each genotype for gel zymography and western blot analysis, as has been demonstrated by previous publications to be adequate for similar animal studies. Matching sections were used between controls and mutants. NIS-Elements BR 3.0 software (Nikon) was used for quantifying the numbers and sizes of neuronal ectopia, the numbers of laminin-positive debris, as well as the numbers of astrocytes. ImageJ software (NIH) was used for quantifying the intensity of immunostainings. In analysis of radial glial cell division, the cleavage plane angle was calculated by determining the angle between the equatorial plate and the ventricular surface. Statistics was performed using Student's $t$ test when comparing two conditions, or one-way ANOVA followed by Tukey's post hoc test when comparing three or more conditions. All data are represented as means ± SEM.

## Acknowledgements

ZH thanks Dr. LF Reichardt for supporting the initial generation of *Ric8a* mutant ES cells, Dr. EA Grove (Chicago) for providing the *Wnt3a-Cre* strain, the late Dr. BA Barres (Stanford) for critiques and input, and Drs. WL Murphy and E Bresnick (UW-Madison) for access to a plate reader and a qPCR machine. We also thank the late Dr. D Oertel (UW-Madison) for critical reading and editing and Dr. L Puglielli (UW-Madison) for critical reading of a previous version of the manuscript. This work was supported by funds from the Departments of Neurology and Neuroscience, UW-Madison, and a Basil O'Connor award from the March of Dimes foundation to ZH

## Additional information

### Funding

| Funder | Grant reference number | Author |
|---|---|---|
| March of Dimes Foundation | Basil O'Connor award | Zhen Huang |

The funders had no role in study design, data collection, and interpretation, or the decision to submit the work for publication.

### Author contributions

Hyo Jun Kwon, Data curation, Formal analysis, Validation, Investigation, Visualization, Methodology; Devi Santhosh, Data curation, Formal analysis, Investigation, Methodology; Zhen Huang, Conceptualization, Data curation, Formal analysis, Supervision, Funding acquisition, Validation, Investigation, Visualization, Methodology, Writing - original draft, Project administration, Writing - review and editing

### Author ORCIDs

Zhen Huang ⓘ https://orcid.org/0000-0002-7997-4144

### Ethics

This study was performed in strict accordance with the recommendations in the Guide for the Care and Use of Laboratory Animals of the National Institutes of Health. All of the animals were handled according to approved Institutional Animal Care and Use Committee (IACUC) protocols (M005345) of the University of Wisconsin – Madison.

Reviewer #1 (Public review): https://doi.org/10.7554/eLife.100446.3.sa1
Reviewer #2 (Public review): https://doi.org/10.7554/eLife.100446.3.sa2
Author response: https://doi.org/10.7554/eLife.100446.3.sa3

## Additional files

### Supplementary files
• MDAR checklist

### Data availability
All data generated or analyzed during this study are included in the manuscript and supporting files.

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
