## [Editor Report · eLife Assessment]

The study describes a link between beta-amyloid monomers, regulation of microglial activity and assembly of neocortex during development. It brings **valuable** findings that have theoretical and practical implications in the field of neuronal migration, neuronal ectopia and type II lissencephaly. Unfortunately, the evidence is **incomplete** and the manuscript would benefit from additional experiments to clarify the relationship between Ric8a and APP and bolster the findings.

---

## [Referee Report · Reviewer #1 (Public review)]

Summary:

The authors want to elucidate which are the mechanisms that regulate the immune response in physiological conditions in cortical development. To achieve this goal, authors used a wide range of mutant mice to analyse the consequences of immune activation in the formation of cortical ectopia in mice.

Strengths:

The authors demonstrated that Abeta monomers are anti-inflammatory and inhibit microglial activation. This is a novel result that demonstrates the physiological role of APP in cortical development.

The current manuscript has been slightly improved by additional experiments and editing of the text (many of the suggestions of the reviewers have not been included). However, the evidence supporting the conclusions of the study is still very weak and inconsistent.

Remaining weaknesses:

-There is no evidence that microglia express Emx1. The paper they referred (Zhang et al., 2014) was performed in adult mice so it is not comparable. Moreover, many other papers are saying that Emx1 is not expressed in microglia. Line 175: change in cytokine expression is not a strong evidence to state that Emx1 is expressed in microglia. Fig. S8: It is not clear whether the staining was performed on neuronal primary culture or cortical section? It is also unclear why there is a partial reduction of Ric8a mRNA levels in Emx1-Ric8a cKO and not a completed deletion?

-NestinCre and Emx1Cre mouse models are targeting the same type of cells in the developing cortex (cortical progenitors, glutamatergic neurons and astrocytes), but with one day difference in expression (Emx1 E9.5 and Nestin E10.5). In fact, previous studies using the same approach (Nestin-Ric8a cKO) found ectopias in the cortex, it is more in line with the results of Emx1-Ric8a cKO shown in the current study. There is no evidence to assume that ric8a deficiency in neural cell lineages is not responsible for basement membrane degradation and ectopia formation in ric8a mutants.

-Additional experiments should be performed to demonstrate that ectopia formation in Emx1-ric8a cKO mutant mice is due to an increase in immune stimulation and not a cell-autonomous effect. Using double cx3cr1-cre and nestin-cre ric8a mutant mice is not an argument to say that elevated immune activation of ric8a deficient microglia during cortical development is responsible for ectopia formation (line 2012-2013)

-The similarities between Ric8a cKO and APP cKO mice are not enough evidence to claim that APP and Ric8a are involved in the same anti-inflammatory pathway in microglia.

-Gel zymography is not the same as Western blot. For the quantification of the relative amount of protein, authors should use western blot and not immunofluorescence intensity as shown in Fig. 5g, h. For western blot, you also load the same amount of protein but you have to normalize your samples with a control protein.

-The graph of BrdU cell distribution in the mutant mice (Fig. S1 F) shows that there are more BrdU cells in bins 5-7 and less in bin 9, indicating an impaired migration of upper cortical neurons in the mutant mice. The authors claimed there are no differences in migration in the result section but the figure showed significant differences. Panels E, F in Fig S2 show the density of Cux1 and Ctip2 cells per area indicating no changes in the generation of upper and lower cortical neurons, but no information about the migration as authors claimed (lines 117-118). (what is the field for Ctip2 counting?). These experiments cannot rule out the possibility of cell-autonomous effect of Ric8a deletion in glutamatergic neurons or radial glial cells.

---

## [Referee Report · Reviewer #2 (Public review)]

Kwon et al. used several conditional KO mice for the deletion of ric8a or app in different cell types. Some of them exhibited pial basement membrane breaches leading to neuronal ectopia in the neocortex.

I am glad to see that the authors performed some of the requested controls.

However, a huge problem with this manuscript which has been highlighted in the reviewer's comments but not corrected by the authors, is the claim that "A novel monomeric amyloid beta-activated signaling pathway regulates brain development". They do not have any proof that Abeta is the activating signal in vivo. Whatever they showed in vitro should be confirmed in vivo to make such a strong claim. The authors even recognized it in their responses to reviewers: "we currently do not have evidence that in the developing cortex Abeta monomers play a role in inhibiting microglia". Therefore, their title is misleading, not supported by the data, and must be changed to reflect accurately the results. Maybe something like "Involvement of microglia in the formation of cortical ectopia".

The abstract is also misleading and must be changed. The abstract is mostly about Abeta, pretending that this is the key part of their findings while they only provide a few in vitro experiments but nothing in vivo.

This is such a bad way to summarize their data. Most of their in vivo data is about Ric8a, then a smaller in vivo part about APP and nothing about Abeta in vivo. But the title "novel monomeric amyloid beta-activated signaling pathway regulates brain development via inhibition of microglia" only mention Abeta. And the Abstract 90% focuses on Abeta.

The first half of the introduction is about Abeta. Why would they focus their paper about Abeta while they basically have only one figure with in vitro data !! This is so deceptive.

It seems that these authors do not fully understand the importance of having their claims supported by solid data.

(1) The authors did not show in vivo data supporting that Abeta monomers are the key players here.

(2) The authors did not show in vivo data supporting the cytokine secretion data provided in vitro in a model system. They claim that it is not technically feasible to extract the extracellular (secreted) fractions of cytokines from an embryonic brain without causing cell lysis and the release of the intracellular pool. But how about RT-qPCR? After all, they showed that the pathway affects the transcription of several cytokines in microglia in vitro.

(3) The authors did not provide a control experiment to show that the insult induced by LPS injection does not induce the phenotype in the ric8a-foxg1-cre mice.

(4) They did not agree to verify the monomer state of their Abeta monomer preparation, even after addition to the culture medium. Abeta have a strong tendency to polymerize. However, because the authors added the requested result with Ab polymers which gave a different outcome. It is OK with me if they don't do it.

(5) The app-cx3cr1-cre +LPS animals show ectopia only in only subsets of mutants and in most cases only in one of the hemispheres. Experiments examining potential changes in MMP9 are therefore difficult and were not done.

I don't mind the inability to perform all the suggestions from the reviewers but it is then necessary to tone down or remove the claims that are not supported by the data.

This kind of issue appears several times later in the text too:

(1) At the end of the introduction "we found that APP and Ric8a form a pathway in microglia that is specifically activated by the monomeric form of Abeta and that this pathway normally inhibits the transcriptional and post-transcriptional expression of immune cytokines by microglia". Data from Abeta and cytokines are only in vitro, so it has to be specified.

(2) Line 282: "Thus, these results indicate that monomeric Abeta possesses a previously unreported anti-inflammatory activity against microglia that strongly inhibits microglial inflammatory activation". Specify in vitro!

(3) Line 322: "We have shown that heightened microglial activation due to mutation in the Abeta monomer-activated APP/Ric8a pathway results in basement membrane degradation and ectopia during cortical development." This is an overstatement. They did not show that Abeta monomers activate the pathway in vivo.

(4) Line 332: "Thus, these results indicate that excessive inflammatory activation of microglia is responsible for ectopia formation in ric8a mutants." This is incorrect. Inhibition of Akt or stat3 does much more than just being pro-inflammatory. This could affect directly migration. The data only show that Akt and/or Stat3 might be involved.

(5) Line 355: "these results indicate this Abeta monomer-regulated anti-inflammatory pathway normally promotes cortical development through suppressing microglial activation and MMP induction.". Another overstatement. There is no proof that Abeta is involved in vivo.

(6) Line 362: "In this article, we have identified a novel microglial anti-inflammatory pathway activated by monomeric Abeta that inhibits microglial cytokine expression and plays essential roles in the normal development of the cerebral cortex". Another overstatement. There is no proof that Abeta is involved in vivo.

(7) Line 365: "this pathway is mediated by APP and the heterotrimeric G protein GEF and molecular chaperone Ric8a in microglia and its activation leads to..." They should mention that its activation was in vitro.

(8) Line 387: "In this study, we have shown that immune over-activation of microglia deficient in a monomeric Ab-regulated pathway results in excessive cortical matrix proteinase activation, leading basement membrane degradation and neuronal ectopia." Another overstatement. There is no support to claim that Abeta is involved in vivo. The immune overactivation was not shown in vivo but only in vitro in a model system that does not even reflect correctly what is happening in vivo due to chronic immune stimulation during in vitro culture.

(9) Line 396: "we have also shown that the anti-inflammatory regulation of microglia in corticogenesis depends on a pathway composed of APP and the heterotrimeric G protein regulator Ric8a." Overstatement. They only showed the anti-inflammatory regulation in vitro and not during corticogenesis.

It is just a matter of rewriting the title, abstract and text in an honest way, in order to make sure that every claim is supported by the data and in some cases acknowledge the weakness of the provided data and describe the multiple interpretations than could be drawn out of them.

---

## [Author Response]

The following is the authors’ response to the original reviews.

**Public Reviews:**

**Reviewer #1 (Public Review):**
Summary:The authors want to elucidate which are the mechanisms that regulate the immune response in physiological conditions in cortical development. To achieve this goal, authors used a wide range of mutant mice to analyse the consequences of immune activation in the formation of cortical ectopia in mice.Strengths:The authors demonstrated that Abeta monomers are anti-inflammatory and inhibit microglial activation. This is a novel result that demonstrates the physiological role of APP in cortical development.Weaknesses:-On the other hand, cortical ectopia has been already described in mouse models in which the amyloid signalling has been disrupted (Herms et al., 2004; Guenette et al., 2006), making the current study less novel.

We agree these previous studies have implicated amyloid precursor protein in cortical ectopia. However, since these studies use whole-body knockouts, they have not implicated the functional roles of specific cell types. Nor have they identified the specific mechanisms underlying the formation of this unique class of cortical ectopia. In contrast, our studies show that the disruption of a novel Abeta-regulated signaling pathway in microglia is the primary cause of ectopia formation in this class of ectopia mutants. This is the first time that microglia have been specifically implicated in the development of cortical ectopia. We further show that elevated MMP activity and resulting cortical basement membrane degradation is the underlying mechanism leading to ectopia formation. This is also the first time that MMP activity and basement membrane degradation (instead of maintenance) have been implicated in cortical ectopia development. As such, our results have provided novel insights into the diverse mechanisms underlying cortical ectopia formation in developmental brain disorders.

One of the molecules analysed is Ric8a, a GTPase activator involved in neuronal development. Authors used the conditional mutant mice Emx1-Ric8a to delete Ric8a from early progenitors and glutamatergic neurons in the pallium. Emx1-Ric8a mutant mice present cortical ectopias and authors attributed this malformation to the increase in inflammatory response due to Ric8a deletion in microglia. Several discordances do not fit this interpretation:- The role of Ric8a in cortical development and function has been already described in several papers, but none of them has been cited in the current manuscript (Kask et al., 2015, 2018; Ruisu et al., 2013; Tonissoo et al., 2006).

We have included reference to the published works on *ric8a* in cortical development in revision.

- Ectopia formation in the cortex has been already described in Nestin-Ric8a cKO mice (Kask et al., 2015). In the current manuscript, authors analyzed the same mutant mice (Nestin-Ric8a), but they did not detect any ectopia. Authors should discuss this discordance.

The expression pattern of *nestin-cre* is known to vary dependent on factors including transgene insertion site, genetic background, and sex. Early studies show, for example, that the *nestin* gene promoter drives *cre* expression in many non-neural tissues in another transgenic line in the FVB/N genetic background (Dubois et al Genesis. 2006 Aug;44(8):355-60. doi: 10.1002/dvg.20226). The specific *nestin-cre* line used in Kask et al 2015 has also been shown to be active in brain microglia and lead to increased microglia pro-inflammatory activity upon breeding to a conditional allele of a cholesterol transporter gene (Karasinska et al., Neurobiol Dis. 2013 Jun:54:445-55; Karasinska et al., J Neurosci. 2009 Mar 18; 29(11): 3579–3589; Takampri et al., Brain Res. 2009 May 13:1270:10-8). These factors may in part underlie the apparent discrepancy. We have now incorporated this discussion into the revision.

- Authors claim that microglia express Emx1, and therefore, Ric8a is deleted in microglia cells. However, the arguments for this assumption are very weak and the evidence suggests that this is not the case. This is an important point considering that authors want to emphasise the role of Ric8a in microglia activation, and therefore, additional experiments should demonstrate that Ric8a is deleted in microglia in Emx1-Ric8a mutant mice.

We have observed altered mRNA expression of several genes in purified microglia cultured from the *emx1-cre* mutants (Supplemental Fig. 8), which indicates that *ric8a* is deleted from microglia and suggests a role of microglial *ric8a* deficiency in ectopia formation. This interpretation is further strengthened by the observation that deletion of *ric8a* from microglia using a microglia-specific *cx3cr1-cre* results in similar ectopia (Fig. 2). We also have other data supporting this interpretation, including data showing induction of the expression of a *cre* reporter in brain microglia by *emx1-cre* and loss of *ric8a* gene expression in microglia cells isolated from *emx1-cre* mutants. These data have now been incorporated into the text and in revised Supplemental Fig. 8 (new panels c-c” & d).

**Reviewer #2 (Public Review):**
Kwon et al. used several conditional KO mice for the deletion of ric8a or app in different cell types. Some of them exhibited pial basement membrane breaches leading to neuronal ectopia in the neocortex.They first investigated ric8a, a Guanine Nucleotide Exchange Factor for Heterotrimeric G Proteins. They observed the above-mentioned phenotype when ric8a is deleted from microglia and neural cells (ric8a-emx1-cre or dual deletion with cre combination cx3cr1 (in microglia) and nestin (in neural cells)) but not in microglia alone or neural cells alone whether it is in CR cells (ric8a-Wnt3a-cre), post-mitotic neurons (nex-cre or dlx5/6-cre), or in progenitors and their progeny (nestin-cre or foxg1-cre). They also show that ric8a KO mutant microglia cells stimulated in vitro by LPS exhibit an increased TNFa, IL6 and IL1b secretion compared to controls (Fig 2). They therefore injected LPS in vivo and observed the neuronal ectopia phenotype in the ric8a-cx3cr1-cre (microglial deletion) cortices at P0 (Fig 2). They suggest that ric8a KO in neuronal cells mimics immune stimulation (but we have no clue how ric8a KO in neural cells would induce immune stimulation).

We agree we do not currently know the precise mechanisms by which mutant microglia are activated in the mutant brain. However, this does not affect the conclusion that deficiency in the Abeta monomer-regulated APP/Ric8a pathway in microglia is the primary cause of cortical ectopia in these mutants, since we have shown that genetic disruption of this pathway in microglia alone by targeting different pathway components, using cell type specific cre, in several different approaches, all results in similar cortical ectopia phenotypes. Regarding the source of the immunogens, there are several possibilities which we plan to investigate in future studies. For example, the clearance of apoptotic cells and associated cellular debris is an important physiological process and deficits in this process have been linked to inflammatory diseases throughout life (Doran et al., Nat Rev Immunol. 2020 Apr;20(4):254-267; Boada-Romero et al., Nat Rev Mol Cell Biol. 2020 Jul;21(7):398-414.). In the embryonic cortex, studies have shown that large numbers of cell death take place starting as early as E12 (Blaschke et al., Development. 1996 Apr;122(4):1165-74; Blaschke et al., J Comp Neurol. 1998 Jun 22;396(1):39-50). Studies have also shown that radial glia and neuronal progenitors play critical roles in the clearance of apoptotic cells and associated cellular debris in the brain (Lu et al., Nat Cell Biol. 2011 Jul 31;13(9):1076-83; Ginisty et al., Stem Cells. 2015 Feb;33(2):515-25; Amaya et al., J Comp Neurol. 2015 Feb 1;523(2):183-96). Moreover, Ric8a-dependent heterotrimeric G proteins have been found to specifically promote the phagocytic activity of both professional and non-professional phagocytic cells (Billings et al., Sci Signal. 2016 Feb 2;9(413):ra14; Preissler et al., Glia. 2015 Feb;63(2):206-15; Pan et al. Dev Cell. 2016 Feb 22;36(4):428-39; Flak et al. J Clin Invest. 2020 Jan 2;130(1):359-373; Zhang et al., Nat Commun. 2023 Sep 14;14(1):5706). Thus, it is probable that the failure to promptly clear up apoptotic cells and debris by mutant radial glia may play a role in triggering mutant microglial activation in *ric8a-emx1-cre* mutants. We have now included these possibilities in the text of the revised manuscript. However, the precise mechanisms remain to be determined in future studies, which, however, do not affect the conclusion of the current study.

The authors then turned their attention on APP. They observed neuronal ectopia into the marginal zone when APP is deleted in microglia (app-cxcr3-cre) + intraperitoneal LPS injection (they did not show it, but we have to assume there would not be a phenotype without the injection of LPS) (Fig 3). (The phenotype is similar but not identical to ric8a-cx3cr1-cre + LPS. They suggest that the reason is because they had to inject 3 times less LPS due to enhanced immune sensitivity in this genetic background but it is only a hypothesis). After in vitro stimulation by LPS, app mutant microglia show a reduced secretion of TNFa and IL6 but not IL1b (this is the opposite to ric8a-cx3cr1-cre microglia cells) while peritoneal macrophages in culture show increased secretion of TNFa, IL1, IL6 and IL23 (fig 3 and Suppl. Fig 9).

We have data showing that that *app-cxcr3-cre* mutants without LPS injection do not show ectopia, which has now been included in the revised supplemental Fig. 9 (new panels c-d). The reason we employ LPS injection is, in the first place, that we do not see a phenotype without the injection. We agree, and have also stated in the text, that the phenotype of the *app* mutants is not as severe as that of the *ric8a* mutant. Besides the low LPS dosage used, we also suggest that other *app* family members may compensate since the ectopia in the *app* family gene mutants reported previously were only observed in *app/aplp1/2* triple knockouts, not even in any of the double knockouts (Herms et al., 2004). We have further clarified this point in the text. These possibilities are also not mutually exclusive. Nonetheless, the results clearly show that microglia specific *app* mutation causes cortical ectopia upon embryonic immune stimulation. They have thus implicated a specifical role of microglial APP in cortical ectopia formation.

The different response of ric8a and app mutant microglia to LPS results from in vitro culturing of microglia. We have shown that, when acutely isolated macrophages are used, these mutants show changes in the same direction (both increased cytokine secretion) (Fig. 4). This demonstrates without culturing app mutant microglial lineage cells indeed behave in the same way as ric8a mutant cells.

The microglia used for analysis in in vitro assays in this study have all been cultured for two weeks before assay. They have thus been under chronic stimulation exposed to dead cells and debris in the culture dish through this period. Previous studies have shown that dependent on the degree of perturbation to the inflammation-regulating pathways, such exposures can differentially affect microglial cytokine expression, sometimes in an opposite direction from expected. For example, under chronic immune stimulation, while the *trem2+/-* microglia, which are heterozygous mutant for the anti-inflammatory Trem2, show elevated pro-inflammatory cytokine expression (as is expected), *trem2-/-* (null) microglia under the same conditions instead not only do not show increases but for some pro-inflammatory cytokines, actually show decreases in expression (Sayed et al.,, Proc Natl Acad Sci U S A. 2018 Oct 2;115(40):10172-10177). In several systems, Ric8a-dependent heterotrimeric G proteins have been shown to act downstream of APP and mediate one of the branches of the signaling activated by APP (Milosch et al., Cell Death Dis. 2014 Aug 28;5(8):e1391; Fogel et al,, Cell Rep. 2014 Jun 12;7(5):1560-1576; Ramaker et al., J Neurosci. 2013 Jun 12;33(24):10165-81; Nishimoto et al., Nature. 1993 Mar 4;362(6415):75-9). Indeed, APP cytoplasmic domain is known to also bind to and signalig through several other proteins including FE65, Mena, and TIP60 (Cao & Sudhof, Science 2001. 293:115-120). It is likely that in microglia Ric8a-dependent heterotrimeric G proteins may also mediate only a subset of the signaling downstream of APP. As such, *app* knockout in microglia may have more severe effects on microglial anti-inflammatory regulation than *ric8a* knockout. As a result, upon chronic immune activation, *app* knockout may lead to a microglial phenotype similar to the *trem2* null mutation phenotype as discussed above, while *ric8a* knockout leads to a phenotype similar to *trem2+/-* phenotype. This may explain the subdued TNF and IL6 secretion by cultured *app* (but not *ric8a*) mutant microglia.

Amyloid beta (Ab) being one of the molecules binding to APP, the authors showed that Ab40 monomers (they did not test Ab40 oligomers) partially inhibit cytokines (TNFa, IL6, IL1b, MCP-1, IL23a, IL10) secretion in vitro by microglia stimulated by LPS but does not affect secretion by microglia from app-cx3cr1-cre (tested for TNFa, IL6, IL1b, IL23a, IL10) (Fig 4, Suppl fig 10) (but still does it in aplp2-cx3cr1-cre) and does not affect secretion by ric8a-cx3cr1-cre microglia (tested for TNFa and IL6 but still suppress IL1b) (Therefore here is another difference between app and ric8a KO microglia).

We have tested the effects of Abeta40 oligomers, which induce instead of suppressing microglial cytokine secretion, and have included the data (new panel j in supplemental Fig. 10). As mentioned above, in several systems, Ric8a-dependent heterotrimeric G proteins have been shown to act downstream of APP and mediate one of the branches of the signaling activated by APP (Milosch et al., Cell Death Dis. 2014 Aug 28;5(8):e1391; Fogel et al,, Cell Rep. 2014 Jun 12;7(5):1560-1576; Ramaker et al., J Neurosci. 2013 Jun 12;33(24):10165-81; Nishimoto et al., Nature. 1993 Mar 4;362(6415):75-9). We assume that this is likely also true in microglia and that Ric8a-dependent heterotrimeric G proteins may mediate a subset and only a subset of the signaling downstream of APP. This may explain the difference in the effects of *app* and *ric8a* knockout mutation in abolishing the anti-inflammatory effects of Abeta monomers on IL-1b vs TNF/IL-6. This difference also suggests that TNF/IL-6 and IL-1b secretion must be regulated by different mechanisms in microglia. Indeed, it is well established in immunology that the secretion of IL1b, but not of TNF or IL6, is regulated by inflammasome-dependent mechanisms (see, for example, Proz & Dixit. Nat Rev Immunol. 2016 Jul;16(7):407-20. doi: 10.1038/nri.2016.58).

The authors injected inhibitors of Akt or Stat3 in the ric8a-emx1-cre cortex and found it suppressed neuronal ectopia (Fig 5, Suppl fig 11). It is not clear whether it suppresses immune stimulation from neuronal cells or immune reaction from microglia cells.

We agree at present the pharmacological approaches we have taken are not able to distinguish these possibilities. However, no matter which is the case, our results still implicate a role of excessive microglial activation in the formation of cortical ectopia and support the conclusion of the study. Thus, while worthwhile of further investigation, this question does not impact the conclusion of the current study. Furthermore, as mentioned, we plan to determine the mechanisms of how *ric8a* mutation in neural cells induces immune activation in future studies. These results will likely enable us to more specifically address this question.

Finally, the authors examined the activities of MMP2 and MMP9 in the developing cortex using gelatin gel zymography. The activity and protein levels of MMP9 but not MMP2 in the ric8a-emx1-cre cortex were claimed significantly increased (Fig 5, Suppl fig 12). Unfortunately, they did not show it in the app-cx3cr1-cre +LPS mouse. They make a connection between ric8a deletion and MMP9 but unfortunately do not make the connection between app deletion and MMP9, which is at the center of the pathway claimed to be important here. Then they injected BB94, a broad-spectrum inhibitor of MMPs or an inhibitor specific for MMP9 and 13. They both significantly suppress the number and the size of the ectopia in ric8a mutants (Fig5).

For all the gelatin gel zymography analysis, we quantify protein concentrations in the cortical lysates using the Bio-Rad Bradford assay kit and load the same amounts of proteins per lane. The results across lanes are all directly comparable. From the quantification, our results clearly show that MMP9 activity levels are increased in the mutants (we have now included whole gel images and quantification in a new supplemental Figure 13). The similar levels of MMP2 in all lanes also provide an internal control further supporting the observation of a specific change in MMP9. For this analysis, we focus on the *ric8a-emx1-cre* mutants since the *app-cx3cr1-cre +LPS* animals show ectopia only in only subsets of mutants and in most cases only in one of the hemispheres. Experiments examining potential changes in MMP9 are therefore unlikely to yield meaningful results. On the other hand, we have clearly shown that the administration of different classes of MMP inhibitors significantly eliminate ectopia in ric8a-emx1-cre mutants. This has strongly implicated a functional contribution of MMPs.

After reading the manuscript, I still do not know how ric8a in neural cells is involved in the immune inhibition. Is it through the control of Ab monomers? In addition, the authors did not show in vivo data supporting that Ab monomers are the key players here. As the authors said, this is not the only APP interactor. Finally, I still do not know how ric8a is linked to APP in microglia in the model.

As detailed above, there are several possibilities including potential deficits in the clearance of apoptotic cells and associated debris that may trigger microglial activation in *ri8ca-emx1-cre* mutants. We will investigate these possibilities in future studies. We have now incorporated these possibilities in the revised text. As for the role of Abeta monomers, we have indicated that we currently do not have evidence that in the developing cortex Abeta monomers play a role in inhibiting microglia. We have also indicated in the manuscript that our conclusion is that a microglial signaling pathway that is activated by Abeta monomers in vitro regulates normal brain development in vivo, not that Abeta monomers themselves regulate brain development. Regarding the link between Ric8a and APP, the reviewer has missed several major lines of supporting evidence. For example, we have shown that Abeta monomers activate a pathway in microglia that inhibits the secretion of several proinflammatory cytokines including TNF, IL-6, IL-10, and IL-23 (Figure 4 and Supplemental Figures 8-10). This inhibition is abolished when either *app* or *ric8a* gene is deleted from microglia. This clearly indicates that *app* and *ric8a* act in the same genetic pathway (the pathway activated by Abeta monomers) in microglia. We also show that this Abeta monomer-activated pathway also inhibits the transcription of several cytokines in microglia. This inhibition is also abolished when either *app* or *ric8a* gene is deleted from microglia. This reinforces the conclusion that *app* and *ric8a* act in the same pathway in microglia. Furthermore, cell type specific deletion of *app* or *ric8a* from microglia in vivo also results in similar phenotypes of cortical ectopia. Together, these results strongly support the conclusion that *app* and *ric8a* act in the same pathway that is activated by Abeta monomers in vitro in microglia. This conclusion is also consistent with published findings that Ric8a dependent heterotrimeric G proteins bind to APP and mediate subsets of APP signaling across different species (Milosch et al., Cell Death Dis. 2014 Aug 28;5(8):e1391; Fogel et al,, Cell Rep. 2014 Jun 12;7(5):1560-1576; Ramaker et al., J Neurosci. 2013 Jun 12;33(24):10165-81; Nishimoto et al., Nature. 1993 Mar 4;362(6415):75-9).

While several of the findings presented in this manuscript are of potential interest, there are a number of shortcomings. Here are some suggestions that could improve the manuscript and help substantiate the conclusions:(1) As the title suggests it, the focus is on Ab and APP functions in microglia. However, the analysis is more focused on ric8a. The connection between ric8a and APP in this study is not investigated, besides the fact that their deletion induces somewhat similar but not identical phenotypes. Showing a similar phenotype is not enough to conclude that they are working on the same pathway. The authors should find a way to make that connection between ric8a and app in the cells investigated here.

As discussed above, the reviewer misses several major lines of evidence showing that APP and Ric8a acts in the same pathway in microglia. Besides the similarity of the ectopia phenotypes, for example, we have shown that Abeta monomers activates a pathway in microglia that inhibits the secretion of several proinflammatory cytokines including TNF, IL-6, IL-10, and IL-23 (Figure 4 and Supplemental Figures 8-11). These inhibitory effects are abolished when either *app* or *ric8a* gene is deleted from microglia. This clearly indicates that *app* and *ric8a* act in the same genetic pathway, a pathway that is activated by Abeta monomers in vitro, in microglia. We also show that this Abeta monomer-activated pathway inhibits the transcription of several cytokine genes in microglia. These effects are again abolished when either *app* or _ric8_a gene is deleted from microglia. This further reinforces the conclusion that *app* and *ric8a* act in the same pathway in microglia. Not only so we also show that the same results are true in macrophages. Thus, these results strongly support the conclusion that *app* and *ric8a* act in the same genetic pathway in microglia. This conclusion is also consistent with published findings that Ric8a dependent heterotrimeric G proteins biochemically bind to APP and mediate subsets of APP signaling across different species (Milosch et al., Cell Death Dis. 2014 Aug 28;5(8):e1391; Fogel et al,, Cell Rep. 2014 Jun 12;7(5):1560-1576; Ramaker et al., J Neurosci. 2013 Jun 12;33(24):10165-81; Nishimoto et al., Nature. 1993 Mar 4;362(6415):75-9).

(2) This would help to show the appearance of breaches in the pial basement membrane leading to neuronal ectopia; to investigate laminin debris, cell identity, Wnt pathway for app-cxcr3-cre + LPS injection as you did for ric8a-emx1-cre.

We have now provided further data on pial basement membrane breaches in the *app-cxcr3-cre + LPS* animals (new panels e-f” in supplemental Fig 9). We have not observed any changes in cell identity or Wnt pathway activity in *ric8a-emx1-cre* mutants. It is thus of limited value to examine potential changes in these areas in the *app-cxcr3-cre + LPS* animals.

(3) As a control, this would help to show that app-cxcr3-cre without the LPS injection does not display the phenotype.

We have the data on *app-cx3cr1-cre* mutants without LPS injection, which show no ectopia. We have now included the data in the revised supplemental Fig. 9 (new panels c-d).

(4) This would help to show the activity and protein levels of MMP9 and MMP2 and perform the rescue experiments with the inhibitors in the app-cx3cr1-cre cortex +LPS.

As discussed above, we focus analysis on the *ric8a-emx1-cre* mutants since *app-cx3cr1-cre +LPS* animals show ectopia in only a subset of mutants and in most cases only in one of the hemispheres. Determining potential changes in MMP9 levels and effects of MMP inhibitors are therefore not likely to yield meaningful data. On the other hand, we have shown that MMP9 levels are increased and administration of different classes of MMP inhibitors eliminate cortical ectopia in *ric8a-emx1-cre* mutants. We have also shown a similar break in the basement membrane in *app-cx3cr1-cre +LPS* animals (new panels e-f” in supplemental Fig 9). These results together strongly implicates a role played by MMPs.

(5) Is MMP9 secreted by microglia cells or neural cells?

Our in situ hybridization data show MMP9 is most highly expressed in a sparse microglia-like cell population in the embryonic cortex, suggesting that microglia may be a major source of MMP9. We have incorporated these data in a new supplemental Fig. 12 (panel a). The precise identity of these cells, however, requires further validation.

(6) The in vitro evidence indicates that one of the multiple APP interactors, ie Ab40 monomers, is less effective in suppressing the expression of some cytokines by microglia cells mutants for ric8a (TNFa and IL6 but still suppress IL1b) or APP (TNFa, IL6, IL1b, IL23a, IL10) when compared to WT. But there are other interactors for APP. In order to support the claim, it seems crucial to have in vivo data to show that Ab40 monomers are the molecules involved in preventing the breach in the pial basement membrane.

As addressed in detail above, we have indicated that our conclusion is that a microglial signaling pathway that is activated by Abeta monomers in vitro regulates normal brain development in vivo, not that Abeta monomers themselves regulate brain development in vivo. We currently do not have evidence that the Abeta monomers play a role in inhibiting microglia during cortical development. There are candidate ligands for the pathway in the developing cortex, the functional study of which, however, is a major undertaking beyond the scope of the current study.

(7) In order to claim that this is specific to Ab40 monomers and not oligomers, it is necessary to show that the Ab40 oligomers do not have the same effect in vitro and in vivo. Also, an assay should be done to show that your Ab preparations are pure monomers or oligomers.

We have tested the effects of Abeta40 oligomers, which induce instead of suppressing microglial cytokine secretion, and have included these data in revision in a new panel j in supplemental Fig. 10. The protocols we use in preparing the monomers and oligomers are standard protocols employed in the field of Alzheimer’s disease research. They have been repeatedly optimized and validated over the past decades.

(8) Most of the cytokine secretion assays used microglia cells in culture. Two results draw my attention. Ric8a deletion increases TNFa and IL6 secretion after LPS stimulation in vitro on microglia cells while app deletion decreases their secretion. Then later, papers show that the decrease in IL1b induced by Ab on microglia cells is prevented by APP deletion but not ric8a deletion. Those two pieces of data suggest that ric8a and APP might not be in the same pathway. In addition, the phenotype from app-cxcr3-cre + LPS injection and ric8a-cxcr3-cre + LPS injection are not exactly the same. It could be due to the level of LPS as the author suggests or it might not be. More experiments are needed to prove they are in the same pathway.

As discussed above, the reviewer misses several major lines of evidence, which strongly support the conclusion that APP and Ric8a act in the same pathway activated by Abeta monomers in microglia (see detailed discussion in point 1 above). The differential response of TNFa/IL-6 of *app* and *ric8a* mutant microglia likely results from chronic immune stimulation during in vitro culturing, which is known to alter microglial cytokine response (see detailed discussion in point 9 below). We have demonstrated that this is indeed the case by showing that, without culturing, acutely isolated *app* and *ric8a* mutant macrophages both display elevated TNFa/IL-6 secretion (Figure 4).

Regarding the different regulation of TNF/IL-6 vs IL-1b by APP and Ric8a, as discussed above, in several systems, Ric8a-dependent heterotrimeric G proteins (which are degraded in ric8a mutant cortices, see new supplemental Fig. 9) have been shown to act downstream of APP and mediate one of the branches of the signaling activated by APP (Milosch et al., Cell Death Dis. 2014 Aug 28;5(8):e1391; Fogel et al,, Cell Rep. 2014 Jun 12;7(5):1560-1576; Ramaker et al., J Neurosci. 2013 Jun 12;33(24):10165-81; Nishimoto et al., Nature. 1993 Mar 4;362(6415):75-9). This is likely also the case in microglia and Ric8a-dependent heterotrimeric G proteins may mediate only a subset of the anti-inflammatory signaling activated by APP. As such, *app*, mutation may abolish all the inhibitory effects of Abeta monomers (both those on TNF/IL-6 and those on IL-1b), but *ric8a* mutation may abolish only a subset only those on TNF/IL-6 but not those on IL-1b. This also suggests that the secretion of TNF/IL-6 and IL-1b must be regulated by different mechanisms in microglia. Indeed, it is well established in immunology that the secretion of IL1b, but not that of TNF or IL6, is regulated by inflammasome-dependent mechanisms (see, for example, Proz & Dixit. Nat Rev Immunol. 2016 Jul;16(7):407-20. doi: 10.1038/nri.2016.58).

(9) How do the authors reconcile the reduced TNFa and IL6 secretion upon stimulation of app mutant microglia with the model where app is attenuating immune response in vivo? Line 213 says that microglia exhibit attenuated immune response following chronic stimulation but I don't know if 3 hours of LPS in vitro is a chronic stimulation.

The reviewer has misunderstood. The microglia used in this study have all been cultured in vitro for approximately two weeks before assay. They have thus been under chronic stimulation exposed to dead cells and debris in the culture dish. Dependent on the degree of perturbation to the inflammation-regulating pathways, such exposures are known to change microglial cytokine expression, sometimes in an opposite direction than expected. For example, under chronic immune stimulation, while the *trem2+/-* microglia, which are heterozygous mutant for the anti-inflammatory Trem2, show elevated pro-inflammatory cytokine expression, *trem2-/-* (null) microglia under the same conditions instead not only do not show increases but for some pro-inflammatory cytokines, actually show decreases in expression (Sayed et al.,, Proc Natl Acad Sci U S A. 2018 Oct 2;115(40):10172-10177). As mentioned, in several systems, Ric8a-dependent heterotrimeric G proteins have also been shown to bind to APP and mediate one of the branches of the signaling activated by APP (Milosch et al., Cell Death Dis. 2014 Aug 28;5(8):e1391; Fogel et al,, Cell Rep. 2014 Jun 12;7(5):1560-1576; Ramaker et al., J Neurosci. 2013 Jun 12;33(24):10165-81; Nishimoto et al., Nature. 1993 Mar 4;362(6415):75-9). Thus, it is likely that in microglia, Ric8a-dependent heterotrimeric G proteins also mediate only a subset of the anti-inflammatory signaling activated by APP. As such, *app* knockout in microglia may have more severe effects than *ric8a* knockout on microglial immune activation, resembling the relationship between *trem2* null vs heterozygous mutation discussed above. As such, it is predicted that chronic immune stimulation such as in vitro culturing will result in attenuated pro-inflammatory cytokine expression in app mutant microglia but elevated cytokine expression in ric8a mutant microglia. This may explain why TNF and IL6 secretion by cultured *app* mutant microglia is subdued, but acutely isolated _a_pp mutant macrophages instead show increased cytokine secretion. The latter may be more representative of the response of app mutant microglia in the absence of chronic stimulation.

(10) Line 119: In their model, the authors suggest that there is a breach in pial basement membrane but that the phenotype is different from the retraction of the radial fibers due to reduced adhesion. So, could the author discuss to what substrate the radial fibers are attached to, in their model where the pial surface is destroyed?

Radial glial endfeet normally bind to the basement membrane via cell surface receptors including the integrin and the dystroglycan protein complexes. We observe free radial glial endfeet at the breach sites, apparently without attachment to any basement membrane. However, we cannot exclude the possibility that there may be residual, broken-off basement membrane components bound to the endfeet that are not detected by the methodology employed.

(11) The authors should show that the increased cytokine secretion observed in vitro is also happening in vivo in ric8a-emx1-cre compared to WT mice and compared to ric8a-nestin-cre mice. Or when app is deleted in microglia (app-cxcr3-cre) + LPS injection compared to WT mice +LPS.

Unfortunately, this is not technically feasible since it is not possible to extract the extracellular (secreted) fractions of cytokines from an embryonic brain without causing cell lysis and the release of the intracellular pool. This, however, does not affect our conclusion that the Abeta monomer-regulated microglia pathway plays a key role in regulates normal brain development since its genetic disruption, by different approaches, clearly results in brain malformation.

(12) The authors injected inhibitors of Akt or Stat3 in the ric8a-emx1-cre cortex and found that it suppressed neuronal ectopia (Fig 5, Suppl fig 11). Does it suppress immune stimulation from neuronal cells or immune reaction from microglia cells?

As discussed above, we agree at present the pharmacological approaches we have taken are not able to distinguish these two possibilities. However, whichever is true, it does not affect our conclusion. Also, we plan to determine the mechanisms of how *ric8a* mutation in neural cells induce immune activation in future studies. These results will likely enable us to adopt specific approaches to address this question.

(13) Fig 5 and Supplementary fig 12: Please show a tubulin loading control in Fig 5i as you did in suppl fig 12 d (gel zymography). Please provide a gel zymography showing side by side Control, mutant and mutant +DM/S3I treatment. The same request for the MMP9 staining. Please provide statistics for control vs mutant for suppl fig 12c and d..

We have now included whole gel zymography images with four control and four mutant individual samples as well as quantification in a new supplemental Fig.13 (panels b-c). This clearly shows increases in MMP9, while the MMP2 levels appear similar between controls and mutants. For all of the experiments of gelatin gel zymography, we quantify protein concentrations in the cortical lysates using the Bio-Rad Bradford assay kit and load the same amounts of proteins per lane. The results across lanes are thus all comparable. The MMP9 staining images for the controls and mutants have also all been taken with the same parameters on the microscope and can be directly compared. The statistics have now been provided as suggested.

(14) Please provide the name and the source of the MMP9/13 inhibitor used in this study.

This inhibitor is MMP-9/MMP-13 inhibitor I (CAS 204140-01-2), from Santa Cruz Biotechnology. This information has been included in revision.

(15) The results show that deletion of ric8a in microglia and neural cells induced pia membrane breaches but no phenotype is apparent in ric8a deletion in microglia or neural cells alone. Then, the results showed that intraperitoneal injection of LPS induced the phenotype in ric8a-cxcr3-cre mutants. It would be beneficial as a control supporting the model to show that the insult induced by LPS injection does not induce the phenotype in the ric8a-foxg1-cre mice.

We agree it may potentially be useful to show that LPS injection does not induce ectopia in *ric8a-foxg1-cre* mice. Unfortunately, since the *ric8a-foxg1-cre* mutation shows no phenotype, we are no longer in possession of this line.

**Recommendations for the authors:**

**Reviewer #1 (Recommendations For The Authors):**
- The information in the abstract and the introduction is only related to app. So, it is very abrupt how authors start the manuscript studying the role of Ric8a, with no information at all about this protein and why the authors want to investigate this role in microglial activation. Later in the manuscript, the authors tried to link Ric8a with app to study the role of app in the inflammatory response and ectopia formation. This link is quite weak as well.

In the last paragraph of the Introduction, we explain the use of the *ric8a* mutant and how it leads to discovery of the Abeta monomer-regulated pathway. We have now improved the writing in revision to make these points especially the link between APP and Ric8a-regulated G proteins more clear. In the Results section, we have also improved the writing on the potential link of Ric8a to APP by highlighting, among others, the fact that *ric8a* and *app* pathway mutants are among a unique group of a few mouse mutants (*ric8a, app/aplp1/2,* and *apbb1/2*) that show cortical ectopia exclusively in the lateral cortex, while all other cortical ectopia mutants also show severe ectopia are at the cortical midline. This suggests that similar mechanisms may underlie the ectopia formation in this small group of mutants.

-In order to validate the mouse model, double immunofluorescence or immunofluorescence+in situ hybridization should be performed to show that microglia express ric8a and that is eliminated in the Emx1-Ric8a mutant mice.

As mentioned above, we have additional lines of evidence showing that *ric8a* is deleted from microglia in *emx1-cre* mutants. This includes data showing induction of the expression of a *cre* reporter in brain microglia by *emx1-cre* and loss of *ric8a* mRNA expression in microglia cells isolated from *emx1-cre* mutants. These data have now been included in revised supplemental Fig. 8.

-In Supplemental Fig. 6, the authors claimed that cell proliferation is normal in Ric8a mutant mice without doing any quantification. They also quantified the angle of mitotic division of progenitors in the ventricular zone, but there are no images for the spindle orientation quantification, and no description of how they did it. In addition, this data is contrary to what has already been published in conditional Ric8a mutant mice (Kask et al., 2015). The Vimentin staining should be improved.

We have provided quantification of cell proliferation (phospho-histone 3 staining at the ventricular surface) in revised supplemental Fig. 6g, which shows no significant differences in the number of positive cells. We have also provided details on the definition of the angle of cleavage plane orientation in revised supplemental Fig. 6h and in the Methods section. We are not sure why the results are different from the other study. We were indeed anticipating deficits in mitotic spindle orientation and spent major efforts in the analysis of this potential deficit. However, based on the data, we could not draw the conclusion.

-Analysis of the MMP9 expression should be done by western blot and not by immunofluorescence. In fact, the MMP9 expression shown in Figure 5g,h, does not correspond with RNA expression shown in gene expression atlas like genepaint or the allen atlas, doubting the specificity of the antibody. The expression of Mmp9 is quite low or absent in the cortex at E13.5-E14.5, making this protein very unlikely to be responsible for laminin degradation during development.

We have performed gelatin gel zymography on MMP2/9, which shows increased MMP9 activity levels in the mutant cortex. This is similar to Western blot analysis (all lanes are loaded with the same amounts of cortical lysates). We have now included whole gel zymography images with four control and four mutant individual samples as well as quantification in a new supplemental Fig.13 (panels b-c). The immunofluorescence staining of MMP9, a different type of analysis, was designed as a complementary approach, the results of which also support the interpretation of increases in MMP9 protein. Regarding MMP9 RNA expression, please also note that MMP9 is secreted, and the protein expression pattern is expected to be different from that of RNA. We have performed wholemount in situ using dissected E13.5 mouse forebrains. Our data (in new supplemental Fig.13a) show that MMP9 mRNA is strongly expressed in a sparse population of cells many of which appear to align along blood vessels. We suspect these are microglial lineage cells populating the embryonic cortex at this stage (see, for example, Squarzoni et al., Cell Rep. 2014 Sep 11;8(5):1271-9. doi: 10.1016/j.celrep.2014.07.042.). Our control in situ using a Tnc5 probe also shows that the MMP9 signal is not a result of nonspecific probe binding. Since the MMP9 expressing cells are very sparse even in the wholemount specimens while most database RNA in situ expression data are obtained using thin sections, we suspect this may be why the signal may have been missed in the databases. As for functional contributions, we agree that we cannot rule roles played by other MMPs. However, based on the ectopia suppression data, our results clearly indicate a critical contribution by MMP9/13.

For MMP9 activity, authors should show the whole membrane with a minimum of three control and three mutant individual samples and with the quantification.- The graphs should be improved, including individual values and titles of the Y axes.

We have included whole membrane zymography images with four control and four mutant individual samples as well as quantification in a new supplemental Fig.13b-c. The graphs have also been improved as suggested.